# Orientated crystallization of FA-based perovskite via hydrogen-bonded polymer network for efficient and stable solar cells

Mubai Li [1,5], Riming Sun[1,5], Jingxi Chang[1], Jingjin Dong[1], Qiushuang Tian[1], Hongze Wang[1], Zihao Li[1], Pinghui Yang[1], Haokun Shi[1], Chao Yang[1], Zichao Wu[1], Renzhi Li[1], Yingguo Yang [2], Aifei Wang[1], Shitong Zhang[3], Fangfang Wang [1] ✉, Wei Huang [1,4] ✉ & Tianshi Qin [1] ✉

Incorporating mixed ion is a frequently used strategy to stabilize black-phase formamidinum lead iodide perovskite for high-efficiency solar cells. However, these devices commonly suffer from photoinduced phase segregation and humidity instability. Herein, we find that the underlying reason is that the mixed halide perovskites generally fail to grow into homogenous and high-crystalline film, due to the multiple pathways of crystal nucleation originating from various intermediate phases in the film-forming process. Therefore, we design a multifunctional fluorinated additive, which restrains the complicated intermediate phases and promotes orientated crystallization of α-phase of perovskite. Furthermore, the additives in-situ polymerize during the perovskite film formation and form a hydrogen-bonded network to stabilize α-phase. Remarkably, the polymerized additives endow a strongly hydrophobic effect to the bare perovskite film against liquid water for 5 min. The unencapsulated devices achieve 24.10% efficiency and maintain >95% of the initial efficiency for 1000 h under continuous sunlight soaking and for 2000 h at air ambient of ~50% humid, respectively.

Formamidinium lead iodide (FAPbI$_3$) based perovskite solar cells (PSCs) have attracted much attention during the past decade and reached a recorded power-conversion efficiency (PCE) of 25.7%[1], owing to the suitable bandgap for a single-junction solar cell[2]. Unfortunately, the photoactive black phase (α-FAPbI$_3$) undergoes a notorious phase transition to the non-perovskite yellow phase (δ-FAPbI$_3$) below a temperature of 150 °C[3]. Tremendous efforts have been devoted to improving the stability of α-FAPbI$_3$. One strategy is introducing additives into the pure-FAPbI$_3$ such as volatile salt (methylammonium chloride (MACl), methylammonium iodide (MAI))[4–7], pseudo-anions

(anion formate (HCOO$^-$), thiocyanate (SCN$^-$))[8,9], and cations (methylenediammonium (MDA$^{2+}$), isopropylammonium (iPAmH$^+$))[10,11], whereas most of those PSCs still need high fabrication temperature to 150 °C. Since the first report of the addition of MAPbBr$_3$ to stabilize the α-FAPbI$_3$[12], mixed cations and anions (MA$^+$, Cs$^+$, Br$^-$, and Cl$^-$)[13] became another commonly used strategy to efficiently form black phase under low annealing temperature of 100 °C. To date, most of the efficiency records for perovskite solar cells have been achieved by mixed-ion FA-based perovskites[7,14–17]. However, small amounts of these ions affect the operational stability of corresponding devices. It has been

[1]Key Laboratory of Flexible Electronics (KLOFE) & Institute of Advanced Materials (IAM), Nanjing Tech University (NanjingTech), Nanjing, Jiangsu 211816, P. R. China. [2]Shanghai Synchrotron Radiation Facility (SSRF), Shanghai Advanced Research Institute, Shanghai Institute of Applied Physics, Chinese Academy of Sciences, 239 Zhangheng Road, Shanghai 201204, P. R. China. [3]State Key Laboratory of Supramolecular Structure and Materials, Jilin University, Changchun, Jilin 130012, P. R. China. [4]Frontiers Science Center for Flexible Electronics, Xi'an Institute of Flexible Electronics (IFE), Xi'an Institute of Biomedical Materials and Engineering, Northwestern Polytechnical University (NPU), Xi'an, Shanxi 710072, P. R. China. [5]These authors contributed equally: Mubai Li, Riming Sun. ✉e-mail: iamffwang2@njtech.edu.cn; iamwhuang@nwpu.edu.cn; iamtsqin@njtech.edu.cn

demonstrated that the mixed-ion perovskites suffer from phase segregation[18] under continuous light illumination[19]. And the formation of pinholes and residual $PbI_2$ in perovskite films caused by volatile cation components seriously affect the device performance under heat or humid conditions[20].

Herein, by using in-situ X-ray diffraction (XRD), in-situ ultraviolet-visible (UV-vis) absorption spectra and in-situ grazing-incidence wide-angle x-ray scattering (GIWAXS), we monitor and provide a deep insight into the intermediate phase, nucleation, and crystallization process of the perovskite films during spin-coating and annealing procedures. We find that the underlying reason for the predicament of the mixed halide perovskite is that it generally fails to grow into homogenous and high-crystalline film, due to the multiple pathways of crystal nucleation originating from various intermediate phases in the film-forming process[21] (Fig. 1a). Therefore, the formation of high-quality and stable perovskite films with ordered crystal orientation and low defect density is essential for low nonradiative energy loss and the

long-term stability of PSCs. To overcome these issues, we successfully designed a multifunctional fluorinated molecule 3-fluoro-4-methoxy-4',4"-bis((4-vinyl benzyl ether) methyl)) triphenylamine (FTPA) ((Fig. 1a, synthesis route and $^1H$ NMR of the molecule as shown in Supplementary Fig. 1, synthesis procedures of FTPA as shown in Supplementary Note 1)) as additive in $FA_{0.95}MA_{0.05}Pb(I_{0.95}Br_{0.05})_3$ perovskite (Abbreviated as FAMA). Four important design criteria for the molecule are shown in Fig. 1a, (i) Triphenylamine acts as the core of the molecule for efficient hole transport and energy level regulation in the bulk and surface of the perovskite film; (ii) Fluorine and oxygen atoms can interact with perovskite to restrain the generation of multi-intermediate phase and promote the orientated crystallization of α-$FAPbI_3$. (iii) The flexible diethyl ether groups act as a solubilizing unit to make FTPA a viscous liquid (inert picture in Supplementary Fig. 1), which facilitate continued interaction between molecule and perovskite during the spin-coating and annealing process. (iv) Vinyl groups endow FTPA with in-situ polymerizing and filling in the grain

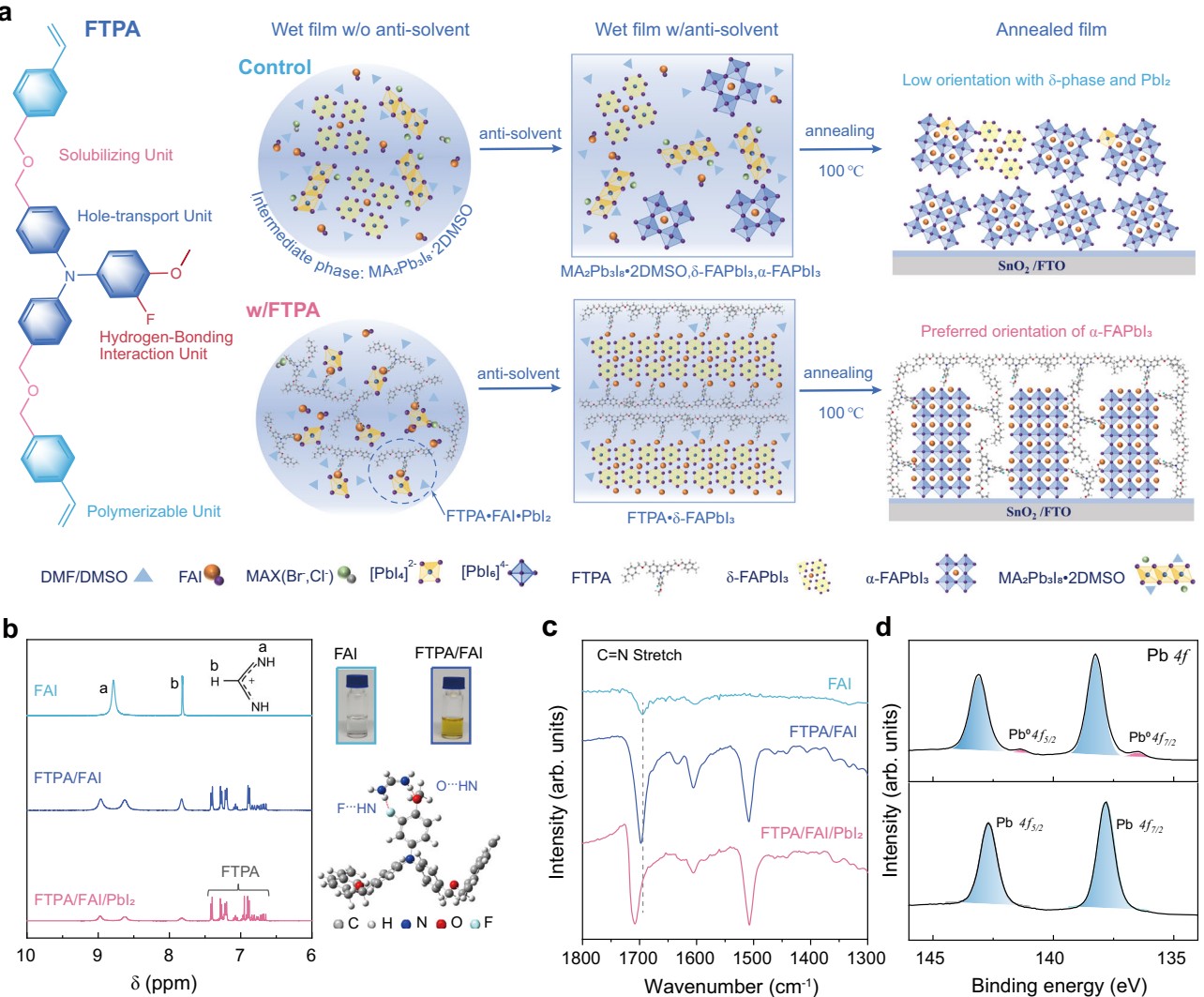

**Fig. 1 | Molecular design and interaction between FTPA and perovskite. a** Molecular structure design of FTPA, and schematic diagram of the possible phase evolution of the nucleation and crystallization of FA-based mixed anion perovskites ($FA_{0.95}MA_{0.05}Pb(I_{0.95}Br_{0.05})_3$) during the film-forming process with (w) or without (w/o) FTPA. In the control perovskite film, the complicated intermediate phases, $MA_2Pb_3I_8\cdot2DMSO$ and δ-$FAPbI_3$, caused two competition pathways of crystal-nucleation as shown in Eqs. (1) and (2), and finally resulted in a low crystal orientation with δ-phase and $PbI_2$. In contrast, perovskite film with FTPA restrained the

formation of the intermediate phases and formed a hydrogen-bonding polymer network in the perovskite films which induced stable and preferred orientation of α-$FAPbI_3$. **b** A comparison of $^1H$ NMR spectra of FAI, FTPA/FAI, and FTPA/FAI/ $PbI_2$. The photos are the FAI solution and FTPA/FAI mixture. Theoretical calculation show that the hydrogen bonds F···HN and O···HN formed between FTPA and FAI have a strength of −29.37 kcal mol⁻¹. **c** FTIR spectra of the FAI, FTPA/FAI, and FTPA/ FAI/$PbI_2$. **d** Pb 4f XPS spectra of the perovskite films of control and with FTPA, respectively.

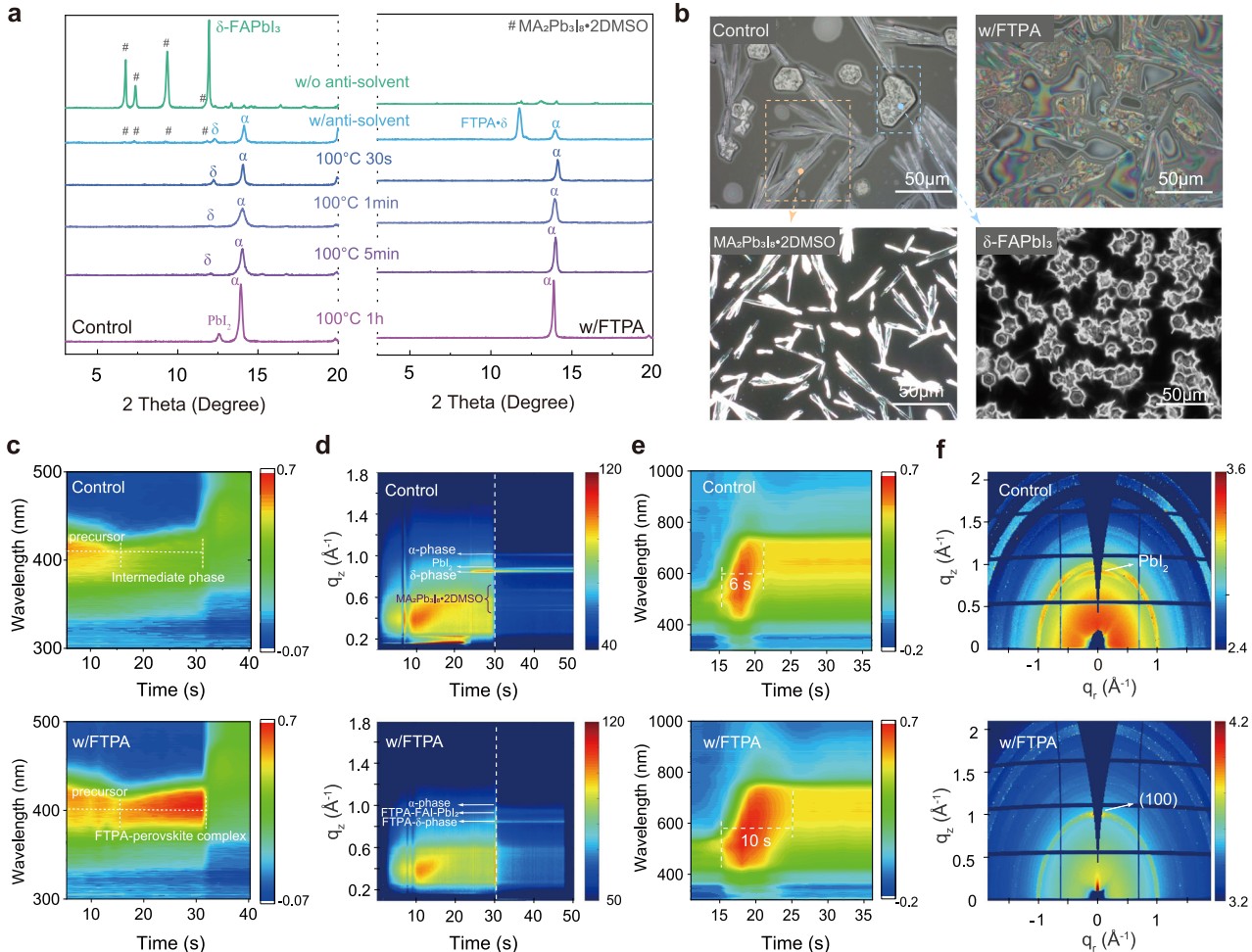

**Fig. 2 | In-situ monitoring of intermediate phase, nucleation, and crystallization process of the control and FTPA based perovskite films during spin-coating and annealing procedures. a** In-situ tracking of X-ray diffraction of the perovskite films during three processes: wet perovskite film without (w/o) anti-solvent during spin-coating, wet perovskite film with (w) anti-solvent during spin-coating, and perovskite films annealed at 100 °C for various times. α and δ symbols indicated α-phase and δ-phase perovskite. **b** Optical microscopy images of the wet perovskite films without antisolvent during the spin-coating process. MA$_2$Pb$_3$I$_8$·2DMSO film was prepared by perovskite of MAI/PbI$_2$ (1:1 mol%) and δ-FAPbI$_3$ was prepared by perovskite of FAI/PbI$_2$ (1:1 mol%). **c** In-situ UV absorption spectra during the spin-coating process. **d** In-situ GIWAXS patterns during spin-coating process. **e** In-situ UV absorption spectra during the initial annealing process at 100 °C. **f** GIWAXS patterns of perovskite films after annealing 1 h.

boundaries of perovskite as a hydrogen-bonding network, which can efficiently stabilize α-FAPbI$_3$ under operational conditions of PSCs. The planar PSCs based on FTPA exhibited a high PCE of 24.10% with photovoltage ($V_{OC}$) of 1.182 V and fill-factor (FF) of 83.45%. Meanwhile, the unencapsulated device exhibited a prominent improvement in operation stability, maintaining the initial efficiency of >95% for 1000 h under continuous sunlight soaking and for 2000 h under air ambient of ~50% humid, respectively. It is noticeable that the perovskite with the polymer network can keep black-phase of FAPbI$_3$ for more than 5 min after immersing the unencapsulated film into the water.

## Results

### Hydrogen-bonded interaction between FTPA and perovskite

The strength of the interaction between the additives and the perovskite precursors determines the intermediate phases and thus plays a key role in managing the crystallization process[13]. FTPA could form hydrogen bonds with charged formamidine (FA$^+$), which were characterized by [1]H NMR (Fig. 1b). In pure deuterated DMSO solution, the resonance signal of protonated ammonium in FAI was at 8.787 ppm (Fig. 1b). With the addition of FTPA, the resonance signal of the ammonium split into two at 8.965 ppm and 8.629 ppm, respectively,

implying that the interaction between FTPA and FA$^+$ led to different chemical environments for the two ammoniums in FA$^+$. These results were consistent with the DFT calculations that the fluorine and oxygen atoms on FTPA were strongly hydrogen-bonded to the ammoniums in FA$^+$. In addition, as shown in the photographs in (Fig. 1b), the transparent FAI solution turned yellow after FTPA was added, which confirmed the formation of the FTPA·FA$^+$ complex in solution. For the sample containing PbI$_2$, the intensity of the ammonium resonance signal decreased but remained split, implying that although there was an interaction between PbI$_2$ and the FTPA·FA$^+$ complex, the interaction between FA$^+$ and FTPA was stronger than that of FA$^+$ with PbI$_2$ (Supplementary Fig. 2 and Supplementary Note 2). We obtained further evidence for the interaction between FTPA and the Pb ions by Fourier Transform Infrared Spectroscopy (FTIR) and X-ray photoelectron spectroscopy (XPS). In both film characterization and device discussed below, the amount of FTPA in all film samples was constant, i. e. the molar ratio of FTPA in the perovskite precursor was 5% (43.13 mg mL$^{-1}$). In FTIR spectra (Fig. 1c and Supplementary Fig. 3), the C = N stretching peak of the pure FAI shifted from 1695 cm$^{-1}$ to 1698 cm$^{-1}$ after addition of FTPA and then further shifted to 1709 cm$^{-1}$ after adding PbI$_2$. As shown in Fig. 1d, XPS spectral profile showed that the main peaks at

143.15 eV (Pb $4f_{5/2}$) and 138.25 (Pb $4f_{7/2}$) eV were shifted toward low binding energy regions (142.70 eV and 137.80 eV, respectively) after the introduction of FTPA in the perovskite, whereas the F $1s$ peak of perovskite with FTPA showed the opposite trend (Supplementary Fig. 4). This could be attributed to the interaction between F group of FTPA and uncoordinated Pb ion. In addition, the small peaks of metallic Pb next to the main peaks in the FTPA perovskite films almost disappeared compared to the control film[22]. Supplementary Fig. 4 showed the I $3d$ XPS spectra assigned as Pb-I chemical species. The I $3d_{3/2}$ and I $3d_{5/2}$ peaks of the FTPA sample at 630.05 eV and 618.55 eV shifted to the lower binding energy region compared to the control (630.45 eV and 618.95 eV). This indicated that the Pb–I bond was weakened due to the interaction with the FTPA-FA$^+$ complex. We also noticed that the C-C=O peak (288.26 eV) in C $1s$ of the control film related to oxygen/moisture was significantly suppressed after the addition of FTPA, demonstrating that FTPA could slow down the degradation of the perovskite layer[7]. The above results indicated the intermediate phase (FTPA-FAI-PbI$_2$) could be formed due to the strong interactions between the FTPA units and perovskite.

## In-situ monitoring of nucleation and crystallization of perovskite film

The interaction between additives and perovskite affects the nucleation and crystallization process of perovskite[23,24] which often occurs rapidly at the stage of spin-coating and initial annealing process[25]. Therefore, in order to understand the transformation process of the intermediate phases into perovskite, we carried out in-situ tracking of XRD (Fig. 2a) to investigate different processes. For the control FAMA perovskites, although only small amounts of MA$^+$ were used in the perovskite precursor, complex intermediate phases, e.g., the solvate phases MA$_2$Pb$_3$I$_8$·2DMSO (2θ = 6.73°, 7.38°, 9.33°) and δ-FAPbI$_3$ (2θ = 11.9°), were formed from the DMSO/DMF solvent in the wet perovskite film without using anti-solvent. The formation of MA$_2$Pb$_3$I$_8$·2DMSO was due to a strong interaction between perovskite precursor and DMSO, which was confirmed by the XRD pattern of the intermediate phase of MA$_2$Pb$_3$I$_8$·2DMSO (Supplementary Fig. 5) and agreed with previous report[26]. After using antisolvent to extract DMF/DMSO, most of the intermediate solvate phase transformed to α-phase of perovskite[9]. However, MA$_2$Pb$_3$I$_8$·2DMSO and δ-FAPbI$_3$ were still present in the film. As the anti-solvent assisted crystallization process[27,28], Cl$^-$ from MACl in the perovskite precursor enter the perovskite lattice, which leads to the diffraction peak of δ-FAPbI$_3$ shifted to high-angle region of 12.3°. The diffraction peak next to δ-FAPbI$_3$ (12.3°) is the residual MA$_2$Pb$_3$I$_8$·2DMSO diffraction peak at 11.87°, which is overlapped with δ-FAPbI$_3$ before using anti-solvent as shown in the enlarged XRD patterns in Supplementary Fig. 5. During annealing process from 30 s to 5 min, the position of the δ-FAPbI$_3$ peak (from 12.3° to 11.9°) and α-FAPbI$_3$ (from 14.15° to 13.9°) gradually shifts to low-angle region owing to the substitution of Cl ions by I ions[5,6]. Noticeably, δ-FAPbI$_3$ was consistently observed during α-FAPbI$_3$ crystallization, which might be due to two competition pathways of crystal-nucleation originating from the intermediate phases as shown below and the schematic diagram is shown in Fig. 1a,

$$MA_2Pb_3I_8 \cdot 2DMSO + 3FAI \xrightarrow{\Delta} 3(\alpha - FAPbI_3) + 2MAI \uparrow + 2DMSO \uparrow \quad (1)$$

$$\delta - FAPbI_3 \xrightarrow{\Delta} \alpha - FAPbI_3 \quad (2)$$

Therefore, it is unfeasible to control the nucleation and crystal growth of the α-FAPbI$_3$ owing to the complex perovskite intermediates. After annealing at 100 °C for 1 h, the peak of α-phase perovskite dominated, but PbI$_2$ (2θ = 12.8°) was observed due to the thermal decomposition of the unstable non-perovskite yellow phase.

With the addition of FTPA (Fig. 2a), the intermediate phases of MA$_2$Pb$_3$I$_8$·2DMSO and δ-FAPbI$_3$ were obviously restrained, and the wet perovskite film without antisolvent exhibited inconspicuous crystalline phase with no discernible diffraction peaks, due to the hydrogen-bonded interaction between FTPA and the perovskite precursors. As shown in $^1$H NMR spectra of Supplementary Fig. 6 and Supplementary Note 3, FTPA could suppress the interaction between MA$^+$ (from MACl and MABr) and PbI$_2$, thereby inhibited the formation of solvate intermediate phases from the source. The photograph of perovskite film containing FTPA was almost colorless compared to the yellow control films, which indicated that less nucleation occurred before antisolvent extraction (Supplementary Fig. 7). After using the anti-solvent, a broad diffraction peak at 11.74° was probably assigned to the intermediate phase of FTPA·δ-FAPbI$_3$, and a small amount of α-FAPbI$_3$ was also observed. With annealing at 100 °C, the FTPA·δ-FAPbI$_3$ intermediate converted to the pure α-FAPbI$_3$ in a short time. Along with increasing the annealing time, the peak of α-phase rose and the width of the half-peak gradually narrowed. After annealing for 1 h, the perovskite film featured only a sharp peak of α-phase (Fig. 2a and Supplementary Fig. 8), which was attributed to a single crystallin pathway during perovskite formation. In addition, the in-situ polymerization of FTPA in the perovskite film is another reason for stabilizing the α-FAPbI$_3$, which would be discussed later. To have a clearly view of the intermediate phase of the perovskite samples, the optical microscopy images of the wet perovskite films without antisolvent were measured, as shown in Fig. 2b and Supplementary Fig. 9. In the control perovskite film, the needles of MA$_2$Pb$_3$I$_8$·2DMSO and blocks of δ-FAPbI$_3$ were observed, which verified the two-competing crystalline intermediate phases discussed above. After the addition of FTPA, the wet perovskite films displayed amorphous features without obvious crystallization owing to hydrogen-bonding interaction of FTPA and the perovskite precursors. This also indicated that the liquid FTPA could continuously restrain the formation of the intermediate phases during the spin-coating process.

The effect of FTPA on perovskite film formation during spin-coating process was explored by in-situ UV-vis absorption spectra and in-situ GIWAXS. In Fig. 2c and Supplementary Fig. 10, it was found that the absorption of perovskite precursors with FTPA was significantly enhanced with the spin-coating time increasing, while the control film was obviously weakened, which probably due to the stronger internal interactions of the FTPA-perovskite complex than that of the solvent intermediate. In-situ GIWAXS patterns and the intensity profiles were shown in Fig. 2d and Supplementary Fig. 11, respectively. In the initial spin-coating stage, the nucleation signals of the wet perovskite film couldn't be identified. After dropping antisolvent onto the control perovskite film at 30 s, the diffraction signals of MA$_2$Pb$_3$I$_8$·2DMSO ($q = 0.47$ Å$^{-1}$, 0.52 Å$^{-1}$, 0.66 Å$^{-1}$), δ-FAPbI$_3$ ($q = 0.85$ Å$^{-1}$)), PbI$_2$ ($q = 0.88$ Å$^{-1}$), and α-FAPbI$_3$ ($q = 1.01$ Å$^{-1}$) were detected[13]. This confirmed that the phase evolution of the crystallization of FA-based mixed anion perovskites was complicated due to the numerous of possible crystalline species and the small differences in their formation energies. With addition of FTPA, the intermediate phase of MA$_2$Pb$_3$I$_8$·2DMSO disappeared. Besides the FTPA·δ-phase and α-phase (weak) signals of FAPbI$_3$, the perovskite film showed a new phase at $q = 0.9523$ Å$^{-1}$, which might be the signal of FTPA·FAI·PbI$_2$ complex.

Furthermore, the formation and crystallization process of perovskite films during annealing process were monitored by in-situ UV-vis absorption spectra and GIWAXS. As shown in Fig. 2e and Supplementary Fig. 12, the absorption intensity of the perovskite films started to increase from about 15 s of annealing, which indicated the formation of perovskite crystals. Moreover, the absorption wavelength range increased from 400–600 nm to 400–800 nm, which corresponded to the transition from the intermediate phase to the α-FAPbI$_3$. The crystalline time of α-phase perovskite in the film with FTPA lasted 10 s, which was longer than that of the control film (6 s). Generally, too fast

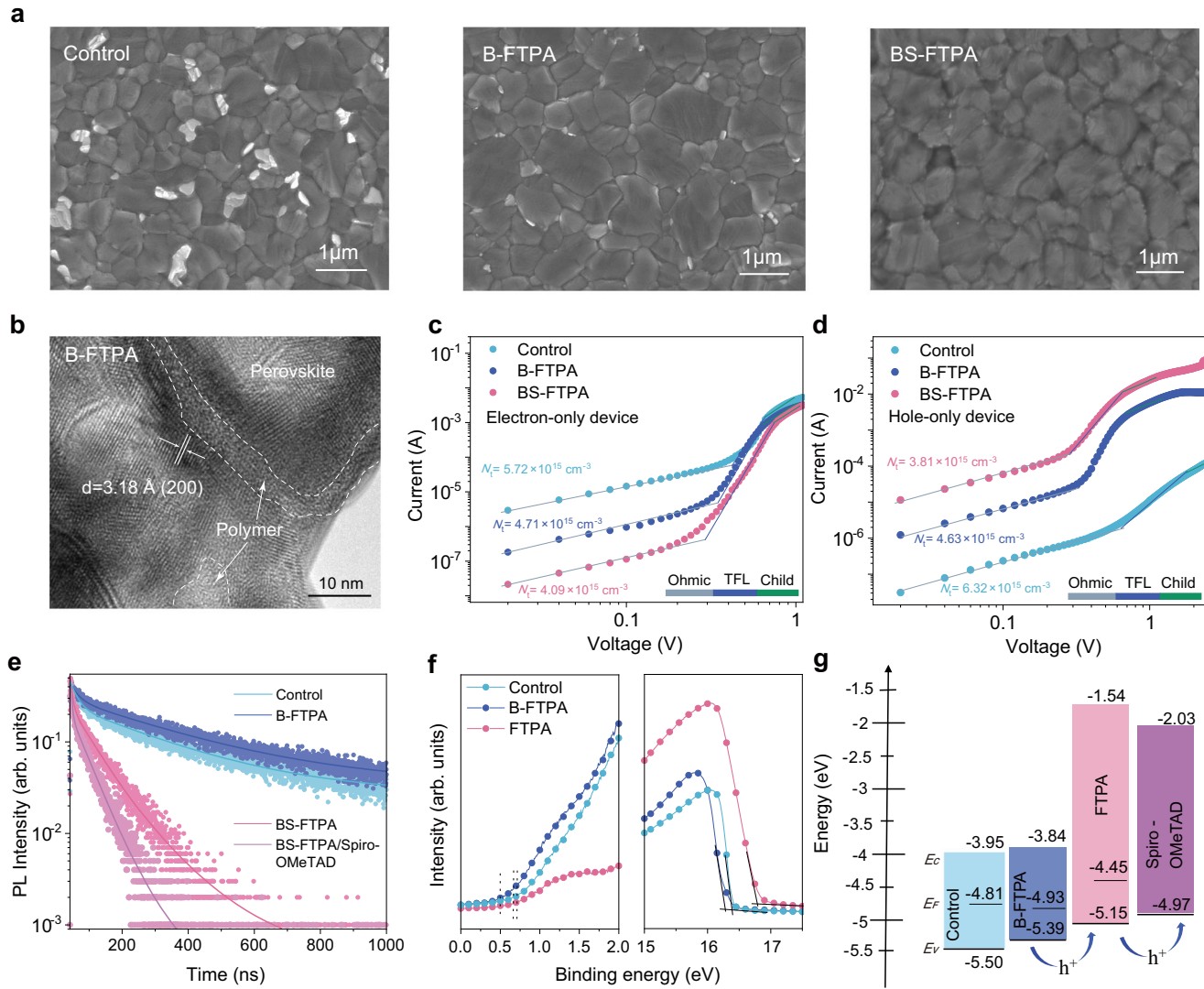

**Fig. 3 | Morphology, carrier extraction and energy level behaviors of the control, B-FTPA and BS-FTPA perovskite films.** B-FTPA is the perovskite film with FTPA in the bulk, and BS-FTPA is the film further spin-coated 2 mg mL$^{-1}$ FTPA and annealed for in-situ polymerizing on the surface of the B-FTPA. **a** Top-view SEM images. **b** HR-TEM image clearly shows the polymerized FTPA surround the grain boundaries of perovskite. *J-V* curves of **c** the electron-only device (FTO/SnO$_2$/perovskite/PCBM/Ag) and **d** the hole-only device (FTO/PEDOT PSS/perovskite/Spiro-OMeTAD/Au) measured by SCLC model. The *J-V* curves can be divided into three regions that are the Ohmic region, trap-filled limit region (TFL), and Child's region. The electron and hole trap density ($N_t$) are shown in Fig. 3c, d, respectively. **e** Time-resolved PL spectra. **f** Valence-band region and photoemission cut off energy of the UPS spectra. **g** Energy level scheme of the devices. Conduction band minimum ($E_C$), Valence band maximum ($E_V$), and Fermi level ($E_F$).

crystallization of perovskite would lead to a high density of defect sites and severe nonradiative recombination[29]. The strong hydrogen-bonding interaction of FTPA-perovskite slowed down the release of cations and anions of perovskite during the annealing process, resulting in an increased energy barrier for perovskite nucleation. The retarded perovskite nucleation and crystal growth are key to high crystal quality. GIWAXS measurements were carried out to probe the crystal orientation of the perovskite films without or with FTPA. As shown in Fig. 2f, GIWAXS pattern exhibited PbI$_2$ peak at $q_z = 0.9$ Å$^{-1}$ in control perovskite film, which was completely eliminated by adding FTPA. Moreover, the control film exhibited diffraction rings of the (100) plane at $q = 1.0$ Å$^{-1}$, implying randomly oriented crystals. Remarkably, the perovskite film with FTPA showed the sharp Bragg spot of (100) plane along the out-of-plane ($q_z$) direction, indicating well-aligned α-FAPbI$_3$ perovskite[25]. As expected, FTPA effectively restrained complicated intermediate phase and retarded perovskite crystallization kinetics, which enabled oriented growth of perovskite films.

## Morphology, carrier extraction and energy level behaviors of perovskite films

We applied FTPA to the bulk of perovskite films (B-FTPA) and further on the B-FTPA film surface (BS-FTPA) to passivate defects, minimize bandgap penalty, and improve charge extraction and transport. FTPA with vinyl groups could be polymerized in-situ via atom transfer radical polymerization (ATRP) in the bulk or on the surface of the perovskite films during the annealing-step. The morphology of the control, B-FTPA, and BS-FTPA perovskite films were studied by scanning electron microscopy (SEM) and atomic force microscope (AFM). As shown in Fig. 3a, white PbI$_2$ phase could be clearly observed in the control film[30]. The B-FTPA perovskite film exhibited the larger grains and the PbI$_2$ phase was disappeared. The BS-FTPA perovskite film showed a completely covered surface by the polymerized FTPA, which was thin enough to discriminate the underlying grain boundaries. The grain-size distributions of these films were displayed in Supplementary Fig. 13. The control film had an average grain-size of ~550 nm, while the B-FTPA and BS-FTPA films exhibited larger size of ~820 nm and

~870 nm, respectively. The surface roughness was reduced from 23.8 nm of the control film to 19.4 nm of the BS-FTPA perovskite, owing to the smoother polymer surface (Supplementary Fig. 14). High-resolution transmission electron microscopy (HR-TEM) images confirmed the in-situ polymerization of FTPA in the bulk of perovskite film. Figure 3b showed that the perovskite grains exhibited noticeable lattice fringes, while the polymerized FTPA with an amorphous morphology existed among or at the edges of the crystalline perovskite grains[31,32]. The lattice spacing of perovskite crystal was determined to be 3.18 Å, which corresponded to the (200) plane of the $FAPbI_3$ crystal cubic phase. The differential scanning calorimeter (DSC) (Supplementary Fig. 15 and Supplementary Note 4) detected that the polymerization temperature of FTPA decreased from 158° to 100° by adding the free radical initiator, which was just the formation temperature of $\alpha$-$FAPbI_3$ crystals. Fourier transform infrared spectroscopy (FTIR) (Supplementary Fig. 16) also confirmed the complete polymerization of FTPA in the perovskite films, according to the vibration peaks ascribed to $RCH = CH_2$ (915 cm$^{-1}$) that all disappeared after annealing at 100°. Therefore, during annealing, FTPA polymerized in the intergranular regions as the grains grew and the hydrogen-bonding interaction retarded perovskite crystallization, which induced orientated growth and good morphology of the perovskite film.

In general, insulating polymers or cross-linked molecules in the perovskite film could passivate the defects but retarded carrier transport. The charge mobility and the trap-state density of electron-only (Fig. 3c) and hole-only devices (Fig. 3d) were characterized by space-charge limited current (SCLC) measurements. Compared to the control device, lower hole (or electron) trap density ($N_t$) obtained in B-FTPA and BS-FTPA could be credited to the passivation of defects in bulk and surface of perovskite films. The calculated electron mobilities of the control, B-FTPA, and BS-FTPA devices were comparable, being $1.98 \times 10^{-3}$ cm$^2$ V$^{-1}$ s$^{-1}$, $1.51 \times 10^{-3}$ cm$^2$ V$^{-1}$ s$^{-1}$, and $1.14 \times 10^{-3}$ cm$^2$ V$^{-1}$ s$^{-1}$, respectively. The corresponding hole mobility of B-FTPA perovskite ($2.74 \times 10^{-4}$ cm$^2$ V$^{-1}$ s$^{-1}$) has increased by an order of magnitude compared to that of the control one ($1.01 \times 10^{-5}$ cm$^2$ V$^{-1}$ s$^{-1}$), due to the excellent hole transport of pure FTPA ($2.97 \times 10^{-2}$ cm$^2$ V$^{-1}$ s$^{-1}$) with triphenylamine as the molecular core (Supplementary Fig. 17 and Supplementary Note 5). The hole mobility of BS-FTPA perovskite further increased to $1.03 \times 10^{-3}$ cm$^2$ V$^{-1}$ s$^{-1}$, which contributed to the balance transport of electrons and holes in perovskite devices[17,33]. In addition, the enhanced hole mobility demonstrated that FTPA promoted the charge transfer between the perovskite grains. The meticulously designed FTPA acting as a connecting bridge of the grains could minimize electrical decoupling or insulation between the perovskite crystals, which could inhibit non-radiative recombination in the perovskite film[31].

To evaluate the dynamics of charge extraction, time-resolved photoluminescence (PL) (Fig. 3e) was measured and the fitted data by a biexponential equation were summarized in Supplementary Table 1. Compared to the control film with $\tau_2 = 282.3$ ns, the B-FTPA perovskite exhibited longer PL lifetime of $\tau_2 = 325.1$ ns, indicating the strongly suppressed non-radiative recombination due to the improved Schottky order of the bulk perovskite by FTPA[34]. Furthermore, BS-FTPA perovskite film showed an average PL lifetime ($\tau_{avg}$) of 50.1 ns, owing to the hole extraction of FTPA capping layer. By using Spiro-OMeTAD as the hole transport layer on S-FTPA film, the PL $\tau_{avg}$ was further reduced to 26.9 ns. The trend of steady-state PL spectra (Supplementary Fig. 18) was consistent with the results of time-resolved PL measurement. Ultraviolet photoelectron spectroscopy (UPS) (Fig. 3f, g) showed a shifted-down Fermi level ($E_F$) from −4.81 eV (control) to −4.93 eV (B-FTPA), indicating a higher p-doping of B-FTPA perovskite relative to the control counterpart[35]. The valence band maximum ($E_V$) of B-FTPA was −5.39 eV and the high occupied molecular orbital (HOMO) of FTPA and Spiro-OMeTAD was −5.15 eV and −4.97 eV, respectively. Therefore, the addition of FTPA to the bulk and surface of the perovskite films led to gradient energy-level alignment, which could promote the hole transport/extraction[16].

## Photovoltaic performance and stability of PSCs

The cross-sectional SEM images of the control and BS-FTPA PSCs were shown in Fig. 4a. The irregular crystals of the control perovskite converted to monolithic grains with indistinguishable grain boundaries after adding FTPA. We believed that FTPA can tune the nucleation and crystal growth processes of the perovskite films by restraining the complex intermediate phases to reduce the grain boundaries, which facilitates carrier transport. We further got insights into the spatial distributions of FTPA in the perovskite films via time-of-flight-secondary-ion mass spectrometry (ToF-SIMS) technique. The cross-section images (Fig. 4a), three-dimensional images (Supplementary Fig. 19) and depth profiles (Supplementary Fig. 20) of ToF-SIMS confirmed the FTPA uniformly distributions in the bulk and surface of perovskite film. Figure 4b showed current density-voltage ($J$-$V$) curves of the control B-FTPA and BS-FTPA PSCs, and the detailed photovoltaic parameters were shown in Table 1. The control cell had a maximum power conversion efficiency (PCE) of 22.48% with a $J_{SC}$ of 24.46 mA cm$^{-2}$, a $V_{OC}$ of 1.143 V and a fill factor of 80.38%. The champion BS-FTPA PSC showed an excellent maximum PCE of 24.10% with a $J_{SC}$ of 24.43 mA cm$^{-2}$, a $V_{OC}$ of 1.182 V and a fill factor of 83.45%. The incident photo-to-electron conversion efficiency (IPCE) measurements (Fig. 4c) showed that the integrated $J_{SC}$ of the control and BS-FTPA PSCs were 23.22 mA cm$^{-2}$ and 23.32 mA cm$^{-2}$, respectively, which well matched the measured $J_{SC}$ under the solar simulator. Notably, the BS-FTPA PSCs showed a high $V_{OC}$ of 1.182 V, which was 93% of the Shockley−Queisser limit $V_{OC}$ (1.27 V) for the absorption threshold of 1.55 eV (Supplementary Fig. 21). The non-radiative recombination losses ($\Delta V_{OC}$ loss) was calculated to be only 0.10 eV[36,37] (Supplementary Fig. 22). Moreover, the Urbach energy ($E_u$) of the BS-FTPA device (Fig. 4d) was 14.5 meV, which was among the lowest values in reported high-performance PSCs, suggesting very low defect density in the BS-FTPA perovskite film[6,38]. Electrochemical impedance spectra (EIS) in Supplementary Fig. 23 and Supplementary Table 2 showed that the BS-FTPA PSCs had a smaller charge-transfer resistance ($R_{ct}$) and a larger carrier recombination resistance ($R_{rec}$) compared to the control device, stemming from the improved charge transport and suppressed non-radiative recombination[39], which was the origin of the higher FF of the BS-FTPA device. The PCE value of the BS-FTPA device was consistent with the stabilized power output near the maximum power point, revealing the operational the photovoltaic device was stable (Supplementary Fig. 24). Statistical analyses of the photovoltaic parameters based on 30 devices revealed good repeatability of the BS-FTPA devices with a higher average PCE of 23.75% compared to the control device (21.76%) (Fig. 4e).

We evaluated the stability of the unencapsulated devices under illumination, relative humidity, and heat conditions according to the Organic Photovoltaic Stability (ISOS) protocols[40]. The B-FTPA and BS-FTPA PSCs showed narrower performance distribution than the control device for six individual cells at different test time. Devices were aged in a thermostats ($23 \pm 2$ °C, Supplementary Fig. 25) at simulated 1-sun illumination under a nitrogen atmosphere (ISOS-LC-1) (Fig. 4f). The PCE of the control device decreased to 59% of its initial PCE after aging 1000 h, while the PCE of BS-FTPA device decreased only 5% of the initial PCE. We also monitored the PCE change of the unencapsulated devices in ambient air of $25 \pm 5$ °C and $50 \pm 10\%$ relative humidity for 2000h (ISOS-D-1) (Fig. 4g). The PCE of the control device dropped to 95% of the initial value after aging for only 150 h, whereas the BS-FTPA device remained 95% of its initial PCE after 2000 h. Generally, in n-i-p type PSCs, Li-doped Spiro-OMeTAD always bring degradation issue to the device, mainly stemming from the hygroscopic nature of Li-TFSI. Water molecules can easily penetrate the perovskite structure, which further degrades the performance of PSCs[41]. The excellent

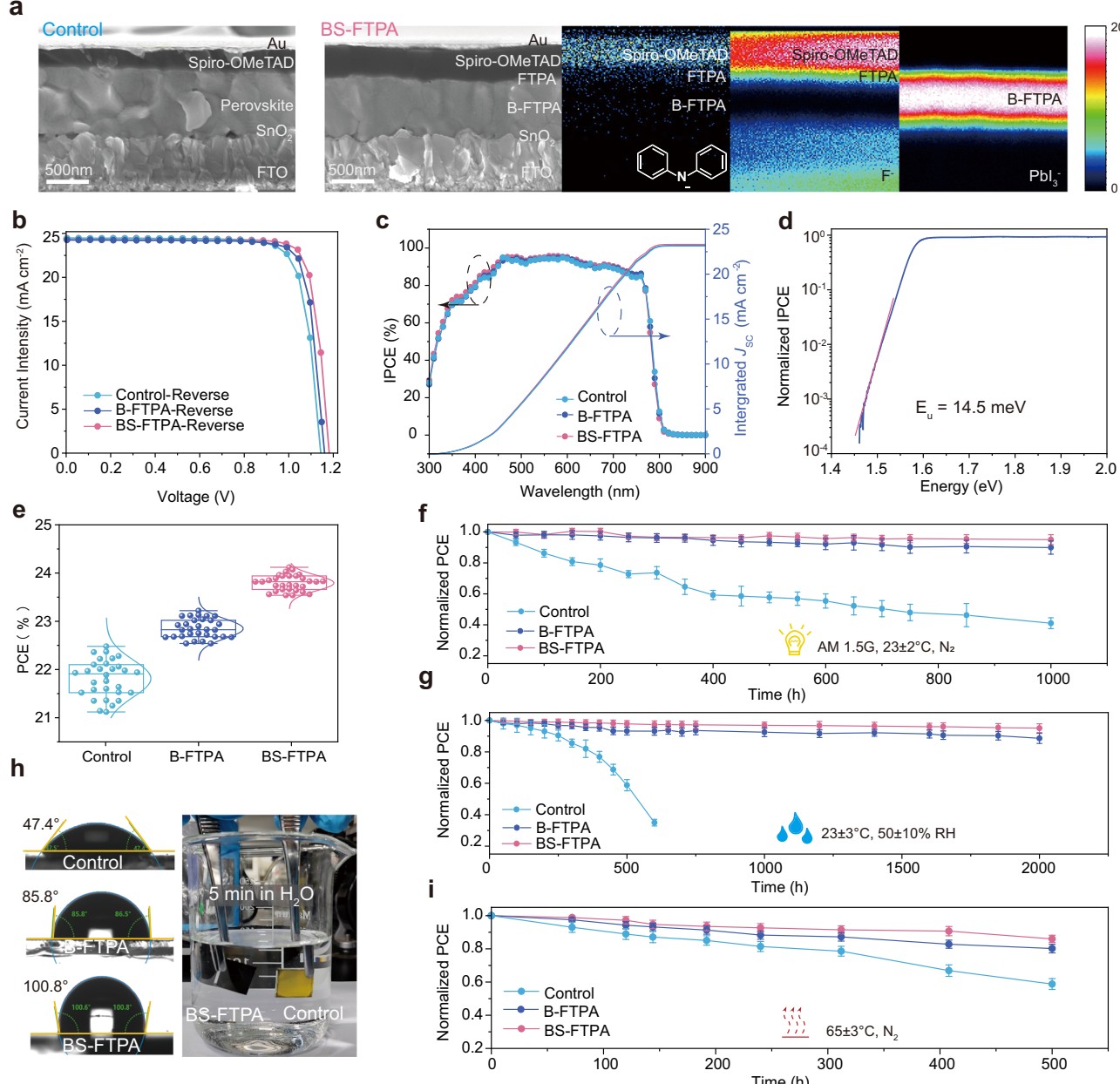

**Fig. 4 | Photovoltaic characteristics and stability of PSCs based on control, B-FTPA and BS-FTPA. a** A cross-sectional SEM images showed the device architectures: FTO/SnO$_2$/perovskite/Spiro-OMeTAD/Au for control, FTO/SnO$_2$/B-FPTA perovskite (with FTPA 43.13 mg mL$^{-1}$ in the bulk)/FTPA (2 mg mL$^{-1}$)/Spiro-OMeTAD/Au for BS-FTPA; ToF-SIMS cross-section images presented the element distribution in BS-FTPA. C$_{12}$H$_{10}$N$^-$ signals attributed to the diphenylamine branches of FTPA and Spiro-OMeTAD; F$^-$ signals corresponding to the Li-TFSI in Spiro-OMeTAD solution, FTPA in bulk and surface of perovskite, and FTO; the PbI$_3^-$ signals corresponding to the perovskite. **b** J-V curves of the champion PSCs. **c** IPCE spectra of the devices integrated over the AM 1.5 G (100 mW cm$^{-2}$) solar spectrum. **d** Semilog plot of IPCE at the absorption onset for BS-FTPA based PSCs, measured using FTPS at J$_{sc}$. An Urbach energy (E$_u$) of 14.5 meV can be obtained from the red line, a sharp absorption edge. **e** Statistical device data based on 30 devices. **f** Device stability of unencapsulated devices under 1-sun illumination at 23 ± 2 °C in a nitrogen atmosphere (ISOS-LC-1). **g** Device stability of unencapsulated devices held at 25 ± 5 °C and 50 ± 10% relative humidity (RH) (ISOS-D-1). **h** Contact angle of perovskite films, and photographs of perovskite films dipped in water, FTO/SnO$_2$/perovskite for control and FTO/SnO$_2$/B-FTPA/FTPA for BS-FTPA. **i** Device stability of unencapsulated devices under 65 ± 3 °C thermal aging (ISOS-T-1). All of the error bars in Fig. 4f, g and i represent the standard deviation for six devices.

humidity stability of BS-FTPA PSCs could be attributed to the hydrophobic effect of polymerized FTPA with fluorinated group on GBs and surface of perovskite, which was confirmed by the significantly increased contact angles of water from 47.4° to 100.8° (Fig. 4h)[42,43]. Surprisingly, the BS-FTPA film could still maintain the black phase of FAPbI$_3$ even after immersing in water for more than 5 min, while the control film without FTPA immediately turned yellow due to the decomposition into PbI$_2$. This confirmed that FTPA could inhibit the penetration of H$_2$O molecules in all directions. The detailed

procedures of the water-immersing test were shown in Supplementary Movie 1. The strong stability of the unsealed perovskite film in water represented a significant progress in stabilizing α-FAPbI$_3$ against moisture and liquid water. We have tracked the XRD patterns to estimate the morphological degradation of the control, B-FTPA and BS-FTPA perovskite films under 85 °C (Supplementary Fig. 26). The control film exhibited obvious deterioration after 500 h manifested by a significant increase of the PbI$_2$ peak, whereas the XRD pattern of the B-FTPA film showed only few of PbI$_2$ peak, and no apparent impurity

**Table 1 | Device parameters of champion PSCs based on different device architecture and perovskite composition via FTPA modification**

| PSC architecture | Perovskite composition | FTPA application | $V_{OC}$ [V] | $J_{SC}$ [mA cm$^{-2}$] | FF [%] | PCE [%] |
|---|---|---|---|---|---|---|
| n-i-p | $FA_{0.95}MA_{0.05}Pb(I_{0.95}Br_{0.05})_3$ | Control[a] | 1.143 | 24.46 | 80.38 | 22.48 |
| | | Bulk[b] | 1.161 | 24.25 | 82.41 | 23.22 |
| | | Bulk/Surface[c] | 1.182 | 24.43 | 83.45 | 24.10 |
| | $Cs_{0.05}FA_{0.85}MA_{0.10}Pb(I_{0.97}Br_{0.03})_3$ | Control[a] | 1.157 | 23.95 | 76.78 | 21.27 |
| | | Bulk[b] | 1.174 | 24.02 | 79.85 | 22.52 |
| | | Bulk/Surface[c] | 1.196 | 24.11 | 80.77 | 23.29 |
| p-i-n | $FA_{0.95}MA_{0.05}Pb(I_{0.95}Br_{0.05})_3$ | Control[d] | 1.031 | 23.01 | 79.06 | 18.75 |
| | | Bulk[e] | 1.126 | 23.22 | 82.09 | 21.46 |
| | $Cs_{0.05}FA_{0.80}MA_{0.15}Pb(I_{0.85}Br_{0.15})_3$ | Control[d] | 1.094 | 23.48 | 79.88 | 20.51 |
| | | Bulk[e] | 1.138 | 23.74 | 81.54 | 22.03 |
| | $Cs_{0.05}FA_{0.80}MA_{0.15}Pb(I_{0.85}Br_{0.15})_3$ | Control[d] | 1.081 | 23.39 | 78.89 | 19.95 |
| | | HTL[f] | 1.156 | 23.93 | 81.27 | 22.48 |

[a]FTO/CBD-SnO$_2$/perovskite/Spiro-OMeTAD/Au;
[b]FTO/CBD-SnO$_2$/perovskite-FTPA/Spiro-OMeTAD/Au;
[c]FTO/CBD-SnO$_2$/perovskite-FTPA/FTPA/Spiro-OMeTAD/Au;
[d]ITO/PTAA/perovskite/PC$_{61}$BM/C$_{60}$/BCP/Ag;
[e]ITO/PTAA/perovskite-FTPA/PC$_{61}$BM/C$_{60}$/BCP/Ag;
[f]ITO/FTPA/perovskite/PC$_{61}$BM/C$_{60}$/BCP/Ag.

peaks were observed in the BS-FTPA film. In addition, the long-term stability of the unencapsulated PSCs at temperature of 65 °C under a nitrogen atmosphere (ISOS-T-1, Fig. 4i) were also examined. After 500 h aging, B-FTPA ad BS-FTPA device stabilized over 80% and 85% of initial efficiency, while the control device only retains 58% of its initial efficiency merely after 500 h. The better thermal stability of FTPA-based PSCs might be attributed to the inhibition of ion migration by the polymer network formed in perovskite film[44].

**The generality of FTPA strategy**

Through investigating the effects of FTPA on perovskite crystallization, energy level modulation, and carrier transport balance, we believe that FTPA should be generally applicable to other cation and halide compositions and different PSC architectures. The corresponding photovoltaic data were summarized in Table 1. We applied FTPA as the perovskite additive in the triplication mixed-halide perovskite system, $Cs_{0.05}FA_{0.85}MA_{0.10}Pb(I_{0.97}Br_{0.03})_3$ (Abbreviated as CsFAMA), and investigated the effect of FTPA on the crystallization kinetics of Cs-containing perovskite system by in-situ XRD, optical microscopy images, SEM images and in-situ UV-vis absorption spectra as shown in Supplementary Fig. 27. In-situ XRD showed that the intermediate phases were also mainly derived from $MA_2Pb_3I_8 \cdot 2DMSO$ and δ-FAPbI$_3$ as the MAFA perovskite system, and the Cs-related intermediate phase δ-CsPbI$_3$ (2θ = 9.7°) was not observed[45]. The addition of FTPA could restrain the intermediate phase and promoted the formation of a better perovskite crystallinity. The optical microscopy images and SEM images also proved that the less intermediate phases during the perovskite formation stage facilitate larger crystal grains of the perovskite films. In-situ UV-vis absorption spectra during spin-coating and annealing process of the perovskite also indicated that the strong internal interactions between the FTPA and perovskite slowed down the crystallization of perovskite, which is essential for the high crystal quality of the treated perovskite. Interestingly, the CsFAMA-perovskite system (5 s) began to crystallize earlier than the FAMA-perovskite system (15 s) after annealing, probably due to the addition of Cs ions promoting the formation of perovskite crystals. PSC performance based on CsFAMA-perovskite in both n-i-p and p-i-n perovskite architectures were improved compared to the reference sample (Supplementary Figs. 28–29 and Supplementary Note 6–7). In addition, benefits from high hole mobility and good energy level alignment with perovskite, we applied FTPA as the dopant-free hole transporting materials (HTM) to alternative the traditional PTAA in p-i-n PSCs[46], and the device exhibited a champion PCE of 22.48% compared to the control device (19.95%) (Supplementary Fig. 30, Supplementary Note 8 and Supplementary Table 3).

## Discussion

Herein, we in-situ monitored and analyzed the intermediate phase, nucleation, and crystallization process of the perovskite films during spin-coating and annealing procedures. We found that the complex intermediate phases were the main reason for disordered crystallization of the mixed halide perovskites, which effected the corresponding photovoltage performance and stability of the PSCs. Based on this understanding, we developed a multifunctional fluorinated additive FTPA, which could suppress the complicated intermediate phase of perovskite and distinctly facilitated the orientated growth of α-FAPbI$_3$. The corresponding PSCs exhibited excellent PCE up to 24.10% owing to several improvements, including balanced charge transport, low defect density, and gradient energy-level alignment. Furthermore, due to the formation of the hydrogen-bonding polymer network in the perovskite film, the stabilized α-FAPbI$_3$ imparted the PSCs with excellent illumination, moisture, and thermal stability. We have opened a successful prospect for the rationalized screening of highly efficient molecular additives for efficient and stable FAPbI$_3$ based PSCs.

## Methods

### Materials

For synthesis of FTPA, all the reagents were purchased from Aladdin and Energy chemical (China). For perovskite fabrication, Formamidinium iodide (FAI, ≥99.5%) and Methylammonium bromide (MABr, ≥99.5%) were purchased from Hangzhou Perovs Optoelectronic Technology Corp (China). Methylammonium chloride (MACl, ≥99.5%), lead bromide (PbBr$_2$, 99.99%), 2,2′,7,7′-tetrakis (N,N-di-pmethoxyphenylamine)−9,9′-spirobifluorene (Spiro-OMeTAD, >99.8%), lithiumbis (trifluoromethanesulfonyl) imide salt(Li-TFSI, ≥99%), FK209-Co(III)-TFSI (≥99%) were purchased from Xi'an Polymer Light Technology Corp (China). Lead iodide (PbI$_2$, 99.99%), azodiisobutyronitrile (AIBN, ≥98%), N,N-dimethylformamide (DMF, >99.5%), dimethyl sulfoxide (DMSO, >99.0%), chlorobenzene (CB, >98.0%), ethyl

acetate (EA, >99.5%), isopropanol (IPA, >99.5%), acetonitrile (ACN, >99.5%), urea (>99.0%), and 4-tert-butyl-pyridine (TBP, >96.0%) were purchased from TCI Shanghai (China). Stannous chloride $SnCl_2 \cdot 2H_2O$ (99.99%), thioglycolic acid (TGA, 98%), and urea (≥99.5%) were purchased from Sigma-Aldrich (USA). All materials were used as received without further modifications.

## Device fabrication
FTO glass was cleaned ultrasonically for 20 min with detergent, deionized (DI) water and ethanol, respectively. And then dried with dry nitrogen and treated with UVO for 15 min. The compact $SnO_2$ film was prepared by chemical bath deposition (CBD)[17]. The perovskite precursor of 1.474 M $FA_{0.95}MA_{0.05}Pb(I_{0.95}Br_{0.05})_3$ was prepared by dissolving 240.76 mg FAI, 706.9 mg $PbI_2$, and 33.76 mg MACl, 8.21 mg MABr, 27.05 mg $PbBr_2$ salts in the 1 mL DMF/DMSO (8:1, v/v) mixed solvent. For the B-FTPA system, the molar ratio of FTPA (with 0.1 mol% AIBN) in the perovskite precursor was 5 % (43.13 mg mL$^{-1}$). The perovskite solution was deposited on $FTO/SnO_2$ by two consecutive spin-coating steps of 1000 rpm for 10 s and 5000 rpm for 30 s, respectively. During the second spin-coating step, 120 μL of EA was deposited onto the film after 20 s. Afterwards, the film was annealed at 100 °C for 1 h. For BS-FTPA system, 2 mg mL$^{-1}$ FTPA was spin-coated on the B-FTPA perovskite film by 3000 rpm for 30 s, then annealed at 100 °C for 10 min. The hole-transport material was prepared by dissolving 30 μL of TBP, 18 μL of Li-TFSI solution (520 mg in 1 mL acetonitrile), 29 μL of FK209-Co(III)-TFSI solution (300 mg in 1 mL acetonitrile) and 73 mg of Spiro-OMeTAD in 1 mL CB. Then spin-coated on perovskite films by 3000 rpm for 30 s. Finally, Au (80 nm) was evaporated as the electrode. Perovskite preparation and device fabrication of the generality use of FTPA described in the Supporting information Figs. 28–30, Supplementary Note 6–8 and Supplementary Table 3.

## Material characterization
The $^1H$ and $^{13}C$-NMR spectra were conducted in $CDCl_3/d_6$-DMSO using a Bruker 400 MHz instrument. MALDI-TOF MS spectra were measured on Waters Q-Tof Premier mass spectrometry. The Differential Scanning Calorimeter (DSC) and Thermogravimetric analysis (TGA) was performed on Shimadzu DSC-60A and Shimadzu DTG-60H at a heating rate of 10 °C min$^{-1}$ under nitrogen atmosphere, respectively. The absorbance spectra were measured by a UV-vis spectrophotometer with an integrating sphere (PerkinElmer, Lambda 950). Theoretical calculations were carried out with a Gaussian 09 D.01 package using b3lyp/6–31 g(d, p) method. $^1H$ and $^{13}C$-NMR spectra, TGA, UV and CV characterization of FTPA were showed in Supplementary Figs. 31–39 and Supplementary Table 4.

## Film characterization
Fourier Transform Infrared Spectroscopy (FTIR) was performed by Thermo-Fisher Nicolet is50 system. X-ray photoelectron spectroscopy (XPS) measurement was carried out on a Thermo-Fisher ESCALAB 250Xi system with a monochromatized Al Kα (for XPS mode) under a pressure of $5.0 \times 10^{-7}$ Pa. Optical microscopy images was carried out using a LV100ND NIKON optical microscopy. In-situ X-ray diffraction (XRD) data were obtained by using a Bruker D8 Advance diffractometer with a high temperature stage, which allows sample to be measured at controlled temperatures. Grazing-incidence wide-angle x-ray scattering (GIWAXS) measurements were performed at beamline BL14B1 of Shanghai Synchrotron Radiation Facility. For ex situ measurement, samples were prepared in the chemistry lab in SSRF with the same procedure described in the film preparation part. The incident angle was set as 0.40°. For the in situ GIWAXS characterization, the spin-coating processes were conducted in a designed nitrogen-filled box, which contains two opposite Kapton windows to permit the transmission of X-rays, and all the processing conditions were kept the

same as the device fabrication process. The GIWAXS data was collected at every 0.5 s and the exposure time is 50 s. The data of GIWAXS measurements were analyzed by Fit-2D and MATLAB, and were displayed in q coordinates. In-situ absorption spectra were measured by an ISAS-HI001 system (Nanjing Ouyi Optoelectronics Technology) consists of light source, detector, a spin-coater or a hot plate as shown in in Supplementary Figs. 10 and 12. The reflectance mode was used for the in-situ absorption measurements by evaporating the Ag layer on the backside of FTO. A Hamamatsu EQ-99-FC:DF001 laser-driven broadband light sources were used for the white light source. Scanning electron microscopy (SEM) was performed on a JEOL5 JSM-7800F operated at 3 kV. The surface morphology of the perovskite film was collected by atomic force microscope (AFM) (Park XE7). High-resolution transmission electron microscopy (HR-TEM) was performed on 7610F-PLUS (JEOL, Japan). Space charge limited current (SCLC) was recorded on Keithley 2450 SMU (Keithley, USA). Time-correlated single photon counting (TCSPC) was performed on Delta-Flex TCSPC system (Horiba, Japan), recorded at 800 nm using excitation with a 520 nm light pulse. Steady-state photoluminescence (PL) was measured via a fluorescent spectrometer, model Hitach F4600 (Hitach, Japan). with excitation at 520 nm. Ultraviolet photoelectron spectroscopy (UPS) was performed by a PHI 5000 VersaProbe III with a He I source (21.22 eV) under an applied negative bias of 9.0 V. The ToF-SIMS was measured by a ToF-SIMS 5–100 instrument (ION-TOF GmbH, Germany). The depth profiling was obtained through a 2 keV Cs sputtering beam raster of $300 \times 300$ μm area.

## Device characterization
Current density-voltage ($J$-$V$) curves were tested on a Keithley 2400 source meter in a solar simulator (Class 3 A, XES-40S3, SAN-EI) after calibrating the light intensity to AM1.5 G one sun (100 mW cm$^{-2}$) in a standard silicon solar cell (QE-B1) calibrated by Newport under AM1.5 G standard light. A black metal mask was used to define the effective active area of the device to be 0.1 cm$^2$. The incident photon-to-electron conversion efficiency (IPCE) measurements were carried out by a QE-R-900AD system (Nanjing Ouyi Optoelectronics Technology). A Hamamatsu S1337-1010BQ silicon diode used for IPCE measurements was calibrated at the National Institute of Metrology, China. The device characterizations of LEDs were carried out on EQE-R-80 system (Nanjing Ouyi Optoelectronics Technology). The Fourier-transform photocurrent spectroscopy (FTPS) was record by a HS-EQE system (Nanjing Ouyi Optoelectronics Technology). It utilized Fourier transform signal processing techniques to enhance and push the detection limit of photocurrent signals. Electrochemical Impedance Spectroscopy (EIS) was measured by an electrochemical workstation (ChenHua, CHI760E, China). The devices are subjected to long-term light stability tests in a nitrogen glove box under white LEDs (1000 W m$^{-2}$ irradiance), tested periodically by a solar simulator (Class 3 A, XES-40S3, SAN-EI). The humidity stability of the devices is tested at a temperature of $25 \pm 10$ °C and a relative humidity of $50 \pm 10$%, measured periodically by a Solar Light Simulator (Class 3 A, XES-40S3, SAN-EI). The optimization data, forward and reverse scans and the detailed parameters of the PSCs are shown in Supplementary Figs. 40–43 and Supplementary Tables 5–8.

## Reporting summary
Further information on research design is available in the Nature Portfolio Reporting Summary linked to this article.

# Data availability
All data generated in this study are provided in the article and Supplementary Information, and the raw data supporting this study are available from the Source Data file. Source data are provided with this paper.

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

## Acknowledgements
This work is supported financially by National Natural Science Foundation of China (52003118 F.W., 62075094 T.Q., 62205143 W.H.); Natural Science Foundation of Jiangsu Province (BK20211537 T.Q.).

## Author contributions
F.W. conceived the idea, designed the experiment, and wrote the manuscript. F.W., T.Q., and W.H. supervised the work. M.L. synthesized and characterized the new material; R.S. and M.L. fabricated the perovskite devices and carried out the PV performance characterizations; J.C. repeated the synthesis of the molecule. J.D. and Y.Y. conducted the GIWAXS measurements, supported by the BL14B1 beamline of the Shanghai Synchrotron Radiation Facility; M.L. and R.S. carried out DSC, XPS, in-situ XRD, TCSPC, UPS and SIMS measurement; R.S. and M.L. carried out SCLC measurement with the assistance of Q.T. and H.W; Z.L. carried out the SEM measurement with the assistance of Z.W.; P.Y. carried out the PLQE and FTPS characterization; M.L. and R.S. carried out the IR measurement and optical microscopy measurement with the assistance of C.Y.; M.L. and R.S. carried out the in-situ absorption measurement with the assistance of H.S.; A.W. carried out the TEM measurement; S.Z. carried out the theoretical calculation. T.Q. and R.L. gave some revise suggestions. All authors discussed the results and commented on the manuscript.

## Competing interests
The authors declare no competing interests.
