## [Peer Review File · Nature Communications]

Orientated crystallization of FA-based perovskite via hydrogen-bonded polymer network for efficient and stable solar cellsREVIEWER COMMENTS

Reviewer #1 (Remarks to the Author):

In this work, the authors designed a multifunctional fluorinated additive, which restrained the complicated intermediate phases and promoted the orientated crystallization of α -FAPbI₃. The additives further in-situ polymerized during the perovskite film formation and formed a hydrogen-bonded network to stabilize the α -FAPbI₃. The unencapsulated devices achieved 24.12% efficiency and maintained >95% of the initial efficiency for 1000h under continuous sunlight soaking and for 2000h at ambient air with ~50% humidity, respectively. This work reports important advances toward high-efficiency and ultra-stable perovskite solar cells. I suggest the following revisions to further strengthen the importance of the work.

1. The manuscript demonstrated the successful application of the FTPA additive in the (FAPbI₃)_{0.95}(MAPbBr₃)_{0.05} perovskite system. How about the more commonly used Cs-containing perovskite? The intermediate in the Cs-containing perovskite system is very different from the Cs-free one. So the effect of FTPA is expected to be different.
2. The FTPA-modified devices showed impressive light and moisture stability. How about the reproducibility? Li-doped Spiro-OMeTAD was used as the HTL in the device, which is one of the major unstable factors for moisture stability, so the data is amazing to me. I suggest the authors provide statistics on the device's stability.
3. For the in-situ characterization, the in-situ GIWAXS experiments are clearly described in the experimental. But the in-situ UV-vis absorption and in-situ XRD are unclear. More detailed information should be supplied.
4. For the IPCE data in Figure 4c, why did the FTPA-modified device show a redshift?

Reviewer #2 (Remarks to the Author):

Li et.al synthesized a new monomer called FTPA and applied it into perovskite bulk film and onto the surface, suppressing the complicated intermediate phase and facilitating the orientated growth of α -FAPbI₃. The corresponding perovskite devices exhibited a PCE up to 24.12% owing to balanced charge transport, low defect density, and gradient energy-level alignment. Furthermore, due to the formation of the hydrogen-bonding polymer network in the perovskite film, the stabilized α -FAPbI₃ imparted the PSCs with excellent illumination and moisture stability. In general, this paper has relatively solid data and impressive results, and thus, it can be considered for acceptance after the major revision. Some comments are listed below for the further improvement:

1. In figure 2a, the samples (with and without FTPA, both) show clear α -FAPbI₃ peaks, but the position of the peak shifts to low-angle region with the increased annealing time. How to explain this phenomenon?

2. The δ -FAPbI₃ peak of control sample without antisolvent (green line) shows a strong δ -FAPbI₃ peak, but the peak position is shifted to high-angle region when using anti-solvent or with annealing. Is there any explanation here? In addition, a diffraction peak next to the δ -FAPbI₃ phase (11.9 degree) exists in figure 2a (control sample, the light blue line). What is this peak assigned to?
3. The authors applied FTPA to the bulk of perovskite films (B-FTPA) and further on the B-FTPA film surface (S-FTPA) to passivate defects, achieving improved device performance. If just applying FTPA onto the control films with bulk passivation, how does the device perform?
4. AIBN was used to facilitate the polymerization process at lower temperature. Is there any residue of AIBN in the final perovskite films?
5. Is FTPA a universal additive? Can it enhance the device performance when applied in the inverted (p-i-n) perovskite solar cells?
6. When referring to the illumination stability, please add the test temperature of the samples. In addition to the light and humidity stability, how about the thermal stability of the B-FTPA and S-FTPA samples?

Reviewer #3 (Remarks to the Author):

In this manuscript, the authors have chosen a very important topic and have focused on the control of crystallization of the FA-based perovskites. The topic area of the work is timely and interesting. The authors show very convincingly that the mixed halide perovskite generally fails to form high-quality film, due to the multiple pathways of crystal nucleation originating from various intermediate phases in the film-forming process. Series of in-situ evidence such as XRD, UV-vis and GIWAXS provide a deep insight into the intermediate phase, nucleation, and crystallization process of the perovskite films during spin-coating and annealing procedures. To overcome the dilemma, the authors ingeniously design a novel multifunctional fluorinated additive and propose a single crystallization pathway approach to further promote the orientated crystallization of α -FAPbI₃ by the hydrogen-bonded polymer network. An impressive efficiency of perovskite solar cells above 24% is obtained with high stability, even the bare perovskite film can against liquid water long periods of time. The rationalized design of the efficient molecular additive in this study could motivate similar work for efficient and stable FAPbI₃ based PSCs.

The analysis in this work is very thorough. The experimental data are comprehensive and credible, which strongly support the conclusion. In summary, I think this work is important and innovative for the field of perovskite solar cells and will be of wide interest to the readers. I strongly recommend publication of this study in Nature Communication. There are a few things that the authors may address to improve the manuscript:

1. As shown in spectra of XPS (in Fig. 1d and Supplementary Fig. 4), to characterize the interaction of the additive with perovskite, the elemental F region are inadequate and the authors should provide the XPS spectrum of the intrinsic FTPA to fully demonstrate the interaction.

2. FTPA is used as an additive in the perovskite precursor to induce crystallization and stabilize the α -phase of FAPbI₃. Whether FTPA affects the perovskite lattice and alters the perovskite band gap is something the authors need to consider.

3. SCLC measurements in Fig. 3c and 3d exhibit the FTPA improved the hole charge transport in the doped perovskite owing to the intrinsic high hole-transport mobility, which contribute to the balance transport of electrons and holes in perovskite devices. And Fig. 3f and 3g also show that FTPA has an aligned energy-level with perovskite as the interfacial hole-transport layer. I am interested in whether the authors have considered the application of FTPA as a hole transport layer to p-i-n structured PSCs as a substitute for the classical PTAA.

Reply to reviewer #1

In this work, the authors designed a multifunctional fluorinated additive, which restrained the complicated intermediate phases and promoted the orientated crystallization of α -FAPbI₃. The additives further in-situ polymerized during the perovskite film formation and formed a hydrogen-bonded network to stabilize the α -FAPbI₃. The unencapsulated devices achieved 24.12% efficiency and maintained >95% of the initial efficiency for 1000h under continuous sunlight soaking and for 2000h at ambient air with ~50% humidity, respectively. This work reports important advances toward high-efficiency and ultra-stable perovskite solar cells. I suggest the following revisions to further strengthen the importance of the work.

General Response: We appreciate the reviewer for reviewing our manuscript and providing constructive comments to improve the work. We carefully considered these comments and the detailed response can be found in the point-to-point response below.

Comment #1: The manuscript demonstrated the successful application of the FTPA additive in the $(\text{FAPbI}_3)_{0.95}(\text{MAPbBr}_3)_{0.05}$ perovskite system. How about the more commonly used Cs-containing perovskite? The intermediate in the Cs-containing perovskite system is very different from the Cs-free one. So the effect of FTPA is expected to be different.

Response: Thanks for the reviewer's important suggestion about the universal use of FTPA. As the reviewer's suggestion, we have applied FTPA as the perovskite additive in the $\text{Cs}_{0.05}\text{FA}_{0.85}\text{MA}_{0.10}\text{Pb}(\text{I}_{0.97}\text{Br}_{0.03})_3$ (Abbreviated as CsFAMA) perovskite system and investigated the effect of FTPA on the crystallization kinetics of Cs-containing perovskite system by in-situ XRD, optical microscopy images, SEM images and in-situ UV-vis absorption spectra as shown in Supplementary Figure 27 as below. And further fabricated the perovskite solar cells using the device architecture of $\text{FTO/CBD-SnO}_2/\text{Cs}_{0.05}\text{FA}_{0.85}\text{MA}_{0.10}\text{Pb}(\text{I}_{0.97}\text{Br}_{0.03})_3\text{-5mol\%FTPA/FTPA}(2 \text{ mg mL}^{-1})/\text{Spiro-OMeTAD/Au}$ as shown in Supplementary Fig. 28 as below.

As shown in Supplementary Fig. 27, we found that in the Cs-containing perovskite system, the intermediate phases ($\text{MA}_2\text{Pb}_3\text{I}_8 \cdot 2\text{DMSO}$ and $\delta\text{-FAPbI}_3$) of the wet perovskite film were the same as in the $(\text{FAPbI}_3)_{0.95}(\text{MAPbBr}_3)_{0.05}$ (Abbreviated as FAMA) perovskite system, and the Cs-related intermediate phase $\delta\text{-CsPbI}_3$ ($2\theta = 9.7^\circ$) was not observed [Wang S., *Adv. Funct. Mater.* **2020**, *30*, 1908343]. After anti-solvent extracting and during the annealing process, the intermediate phase still maintained, which effects the crystallization of perovskite. With the addition of FTPA, the intermediate phase was restrained and formed a pure δ -phase after using anti-solvent. Finally, after 1h annealing, the FTPA based perovskite film featured a better crystallinity with higher α -phase diffraction peak. The optical microscopy images and SEM images also proved that the less intermediate phases during the perovskite formation stage facilitate larger crystal grains of the perovskite films. In-situ UV-vis absorption spectra during spin-coating and annealing process of the perovskite also indicated that the strong internal interactions between the FTPA and perovskite slowed down the crystallization of perovskite, which is essential for the high crystal quality of the treated perovskite. Interestingly, the CsFAMA-perovskite system (5s) began to crystallize earlier than the FAMA-perovskite system (15s) after annealing, probably due to the addition of Cs ions promoting the formation of perovskite crystals.

As shown in Supplementary Fig. 28, by using the FTPA as additives in CsFAMA-perovskite, PSCs showed a PCE of 22.52%. Further using FTPA as the interface layer, we got a PCE of 23.29%

with the V_{OC} of 1.196 V, J_{sc} of 24.11 mA cm⁻² and a FF of 80.77%, while the control PSCs only showed a lower PCE of 21.27% with the V_{OC} of 1.157 V, J_{sc} of 23.96 mA cm⁻² and a FF of 76.78%.

Changes in the Supporting information.

Supplementary Fig. 27. In-situ monitoring of intermediate phase, nucleation, and crystallization process of the CsFAMA and FTPA based perovskite films during spin-coating and annealing procedures. **a**, In-situ tracking of X-ray diffraction of the perovskite films during three processes: wet perovskite film without anti-solvent during spin-coating, wet perovskite film with anti-solvent during spin-coating, and perovskite films annealed at 120 °C for various times. **b**, Optical microscopy images of the wet perovskite films without antisolvent during the spin-coating process. The bar is 50 μ m **c**, Top-view SEM images and distribution histogram of crystal grain sizes of the perovskite thin films. The bar is 200 nm **d**, The in-situ absorption spectra during the spin-coating process. **e**, In-situ absorption spectra during the initial annealing process at 120 °C.

Supplementary Fig. 28. Optimal J - V curves of (a) $\text{Cs}_{0.05}\text{FA}_{0.85}\text{MA}_{0.10}\text{Pb}(\text{I}_{0.97}\text{Br}_{0.03})_3$, (b) B-FTP and (c) BS-FTP PSCs in the reverse scan (RS) direction, respectively.

Based on the comments of the three reviewers, we have added the following discussion of the generality of FTPA in PSCs.

Changes in the manuscript:

“The generality of FTPA strategy

Through investigating the effects of FTPA on perovskite crystallization, energy level modulation, and carrier transport balance, we believe that FTPA should be generally applicable to other cation and halide compositions and different PSC architectures. The corresponding photovoltaic data were summarized in Table 1. We applied FTPA as the perovskite additive in the triplication mixed-halide perovskite system, $\text{Cs}_{0.05}\text{FA}_{0.85}\text{MA}_{0.10}\text{Pb}(\text{I}_{0.97}\text{Br}_{0.03})_3$ (Abbreviated as CsFAMA), and investigated the effect of FTPA on the crystallization kinetics of Cs-containing perovskite system by in-situ XRD, optical microscopy images, SEM images and in-situ UV-vis absorption spectra as shown in Supplementary Fig. 27. In-situ XRD showed that the intermediate phases were also mainly derived from $\text{MA}_2\text{Pb}_3\text{I}_8 \cdot 2\text{DMSO}$ and $\delta\text{-FAPbI}_3$ as the MAFA perovskite system, and the Cs-related intermediate phase $\delta\text{-CsPbI}_3$ ($2\theta = 9.7^\circ$) was not observed.⁴⁵ The addition of FTPA could restrain the intermediate phase and promoted the formation of a better perovskite crystallinity. The optical microscopy images and SEM images also proved that the less intermediate phases during the perovskite formation stage facilitate larger crystal grains of the perovskite films. In-situ UV-vis

absorption spectra during spin-coating and annealing process of the perovskite also indicated that the strong internal interactions between the FTPA and perovskite slowed down the crystallization of perovskite, which is essential for the high crystal quality of the treated perovskite. Interestingly, the CsFAMA-perovskite system (5s) began to crystallize earlier than the FAMA-perovskite system (15s) after annealing, probably due to the addition of Cs ions promoting the formation of perovskite crystals. PSC performance based on CsFAMA-perovskite in both n-i-p and p-i-n perovskite architectures were improved compared to the reference sample (Supplementary Fig. 28-29). In addition, benefits from high hole mobility and good energy level alignment with perovskite, we applied FTPA as the dopant-free hole transporting materials (HTM) to alternative the traditional PTAA in p-i-n PSCs,⁴⁶ and the device exhibited a champion PCE of 22.48% compared to the control device (19.94%) (Supplementary Fig. 30 and Supplementary table 3).”

Table 1. Device parameters of champion PSCs based on different device architecture and perovskite composition via FTPA modification.

PSC Architecture	Perovskite Composition	FTPA Application	V _{oc} [V]	J _{sc} [mA cm ⁻²]	FF [%]	PCE [%]
n-i-p	FA _{0.95} MA _{0.05} Pb(I _{0.95} Br _{0.05}) ₃	Control ^{a)}	1.143	24.46	80.38	22.48
		Bulk ^{b)}	1.161	24.25	82.41	23.22
		Bulk/Surface ^{c)}	1.182	24.43	83.45	24.10
	Cs _{0.05} FA _{0.85} MA _{0.10} Pb(I _{0.97} Br _{0.03}) ₃	Control ^{a)}	1.157	23.95	76.78	21.27
		Bulk ^{b)}	1.174	24.02	79.85	22.52
		Bulk/Surface ^{c)}	1.196	24.11	80.77	23.29
p-i-n	FA _{0.95} MA _{0.05} Pb(I _{0.95} Br _{0.05}) ₃	Control ^{d)}	1.031	23.01	79.06	18.75
		Bulk ^{e)}	1.126	23.22	82.09	21.46
	Cs _{0.05} FA _{0.80} MA _{0.15} Pb(I _{0.85} Br _{0.15}) ₃	Control ^{d)}	1.094	23.48	79.88	20.51
		Bulk ^{e)}	1.138	23.74	81.54	22.03
	Cs _{0.05} FA _{0.80} MA _{0.15} Pb(I _{0.85} Br _{0.15}) ₃	Control ^{f)}	1.081	23.39	78.89	19.94
		HTL ^{g)}	1.156	23.93	81.27	22.48

a) FTO/CBD-SnO₂/perovskite/Spiro-OMeTAD/Au; b) FTO/CBD-SnO₂/perovskite-FTPA/Spiro-OMeTAD/Au; c) FTO/CBD-SnO₂/perovskite-FTPA/FTPA/Spiro-OMeTAD/Au; d) ITO/PTAA/perovskite/PC₆₁BM/C₆₀/BCP/Ag; e) ITO/PTAA/perovskite-FTPA/PC₆₁BM/C₆₀/BCP/Ag; f) ITO/PTAA/perovskite/PC₆₁BM/C₆₀/BCP/Ag; g) ITO/FTPA/perovskite/PC₆₁BM/C₆₀/BCP/Ag.

Comment #2: The FTPA-modified devices showed impressive light and moisture stability. How about the reproducibility? Li-doped Spiro-OMeTAD was used as the HTL in the device, which is one of the major unstable factors for moisture stability, so the data is amazing to me. I suggest the authors provide statistics on the device's stability.

Response: Many thanks for the reviewer's suggestion. We absolutely agree the reviewer's opinion. Generally, in n-i-p type PSCs, Li-doped Spiro-OMeTAD always bring degradation issue to the device, mainly stemming from the hygroscopic nature of Li-TFSI. Water molecules can easily penetrate the perovskite structure, which further degrades the performance of PSCs. (C.C. Boyd, et al., *Chem. Rev.* **2019**, *119*, 3418-3451). The excellent humidity stability of BS-FTPA PSCs could be attributed to the hydrophobic effect of polymerized FTPA with fluorinated group on GBs and surface of perovskite, which was confirmed by the significantly increased contact angles of water from 47.4° to 100.8° (Fig. 4h).

During the stability evaluation, we have measured six parallel individual cells to verify the reproducibility of the PSCs. As the reviewer's suggestion, we have provided statistics on the device's stability in the revised manuscript.

Changes in the manuscript:

“We evaluated the stability of the unencapsulated devices under illumination, relative humidity, and heat conditions according to the Organic Photovoltaic Stability (ISOS) protocols.⁴⁰ The B-FTPA and BS-FTPA PSCs showed narrower performance distribution than the control device for six individual cells at different test time. Devices were aged in a thermostats (23±2 °C, Supplementary Fig. 25) at simulated 1-sun illumination under a nitrogen atmosphere (ISOS-LC-1) (Fig. 4f). The PCE of the control device decreased to 59% of its initial PCE after aging 1000h, while the PCE of BS-FTPA device decreased only 5% of the initial PCE. We also monitored the PCE change of the unencapsulated devices in ambient air of 25±5 °C and 50±10% relative humidity for 2000h (ISOS-D-1) (Fig. 4g). The PCE of the control device dropped to 95% of the initial value after aging for only 150 hours, whereas the BS-FTPA device remained 95% of its initial PCE after 2000 hours. Generally, in n-i-p type PSCs, Li-doped Spiro-OMeTAD always bring degradation issue to the device, mainly stemming from the hygroscopic nature of Li-TFSI. Water molecules can easily penetrate the perovskite structure, which further degrades the performance of PSCs.⁴¹ The excellent humidity stability of BS-FTPA PSCs could

be attributed to the hydrophobic effect of polymerized FTPA with fluorinated group on GBs and surface of perovskite, which was confirmed by the significantly increased contact angles of water from 47.4° to 100.8° (Fig. 4h)^{42,43}

Fig. 4 f, Device stability of unencapsulated devices under 1-sun illumination at $23 \pm 2^\circ\text{C}$ in a nitrogen atmosphere (ISOS-LC-1). **g,** Device stability of unencapsulated devices held at $25 \pm 5^\circ\text{C}$ and $50 \pm 10\%$ relative humidity (ISOS-D-1). **i,** Device stability of unencapsulated devices under $65 \pm 3^\circ\text{C}$ thermal aging (ISOS-T-1). All of the error bars represent the standard deviation for six devices.

Comment #3: For the in-situ characterization, the in-situ GIWAXS experiments are clearly described in the experimental. But the in-situ UV-vis absorption and in-situ XRD are unclear. More detailed information should be supplied.

Response: Thanks for the important comment. As shown in Supplementary Fig. 10 and Supplementary Fig. 12, the in-situ absorption spectra were measured by an ISAS-HI001 system (Nanjing Ouyi Optoelectronics Technology) consists of light source, detector, a spin-coater or a hot plate. The reflectance mode was used for the in-situ absorption measurements by evaporating the Ag layer on the backside of FTO. In-situ X-ray diffraction (XRD) data were obtained by using a Bruker D8 Advance diffractometer with a high temperature stage, which allows sample to be measured at controlled temperatures.

Changes in the experiment section of manuscript:

Change 1: In-situ absorption spectra were measured by an ISAS-HI001 system (Nanjing Ouyi Optoelectronics Technology) consists of light source, detector, a spin-coater or a hot plate as shown in Supplementary Fig. 10 and Supplementary Fig. 12. The reflectance mode was used for the in-situ absorption measurements by evaporating the Ag layer on the backside of FTO.

Change 2: In-situ X-ray diffraction (XRD) data were obtained by using a Bruker D8 Advance diffractometer with a high temperature stage, which allows sample to be measured at controlled temperatures.

Supplementary Fig. 10.

Supplementary Fig. 12.

Supplementary Fig. 10. (c) Schematic of the in-situ UV-vis absorption measurement consisting of light source, detector, and a spin-coater.

Supplementary Fig. 12. (c) Schematic of the in-situ UV-vis absorption measurement consisting of light source, detector, and a hot plate.

Comment #4: For the IPCE data in Figure 4c, why did the FTPA-modified device show a redshift?

Response: We sincerely appreciate review's important comment. We measured the UV absorption of the perovskite film and confirmed that the perovskite band gap was not changed at 1.55eV after addition of FTPA compared to the control film as shown in Supplementary Fig. 21. The small redshift of IPCE might come from errors in the instrument measurements. Therefore, we refabricated and retested the FTPA-modified device (BS-FTPA) and replaced it with the accurate data as shown in Figure 4c.

Supplementary Fig. 21. UV-vis absorption spectra of the Control, B-FTPA and BS-FTPA perovskite films.

Fig. 4c IPCE spectra of the devices integrated over the AM 1.5G (100Mw cm^{-2}) solar spectrum.

Reply to reviewer #2

Li et.al synthesized a new monomer called FTPA and applied it into perovskite bulk film and onto the surface, suppressing the complicated intermediate phase and facilitating the orientated growth of α -FAPbI₃. The corresponding perovskite devices exhibited a PCE up to 24.12% owing to balanced charge transport, low defect density, and gradient energy-level alignment. Furthermore, due to the formation of the hydrogen-bonding polymer network in the perovskite film, the stabilized α -FAPbI₃ imparted the PSCs with excellent illumination and moisture stability. In general, this paper has relatively solid data and impressive results, and thus, it can be considered for acceptance after the major revision.

General Response: We thank the reviewer for the nice comment and well appreciating the importance of our work. We carefully considered these comments and the detailed response can be found in the point-to-point response below.

Comment #1: In Figure 2a, the samples (with and without FTPA, both) show clear α -FAPbI₃ peaks, but the position of the peak shifts to low-angle region with the increased annealing time. How to explain this phenomenon?

Response: Many thanks for the reviewer's comment. This is an important question for our understanding of the crystalline evolution of FA-based perovskite. In our perovskite system, the perovskite precursor of 1.474M (FAPbI₃)_{0.95}(MAPbBr₃)_{0.05} was prepared by dissolving 240.76 mg FAI, 706.9 mg PbI₂, and 33.76 mg MAcl, 8.21 mg MABr, 27.05 mg PbBr₂ salts in the 1 mL DMF/DMSO (8:1, v/v) mixed solvent. In the composition of FA-based perovskite, MAcl is a general additive, which used to assist phase transfer from δ -phase to α -phase of FAPbI₃ perovskite. (M. Kim, et al., *Joule*, **2019**, 3, 2179; Ye, F. et al. *Adv. Mater.*, **2021**, 33, 2007126). There is a large initial amount of Cl present in the wet perovskite before annealing. During annealing process, MAcl will decompose into MA ion and FACl, and then evaporate from the film, and the halide mixture composition does not change. As the reviewer observed, the position of the α -FAPbI₃ peak shifts to low-angle region as the increased annealing time in the in-situ XRD measurement. This phenomenon is attributed to the substitution of Cl ions by I ions during the annealing process, which is also observed in Kim et. al.'s report. (M. Kim, et al., *Joule*, **2019**, 3, 2179).

Figure 2 a, In-situ tracking of X-ray diffraction of the perovskite films during three processes: wet perovskite film without anti-solvent during spin-coating, wet perovskite film with anti-solvent during spin-coating, and perovskite films annealed at 100 °C for various times.

Comment #2: The δ -FAPbI₃ peak of control sample without antisolvent (green line) shows a strong δ -FAPbI₃ peak, but the peak position is shifted to high-angle region when using anti-solvent or with annealing. Is there any explanation here? In addition, a diffraction peak next to the δ -FAPbI₃ phase (11.9 degree) exists in figure 2a (control sample, the light blue line). What is this peak assigned to?

Response: We sincerely appreciate the review's valuable comment and helpful suggestion. We have carefully compared and analyzed the XRD data. In the wet perovskite film, before using anti-solvent, the diffraction peak δ -FAPbI₃ is at 11.9°. As the anti-solvent assisted crystallization process [Jeon, N.J. et al. *Nat. Mater.* **13**, 897-903 (2014); Xiao, M. et al. *Angew. Chem. Int. Ed.* **53**, 9898-9903 (2014).], Cl⁻ from MACl in the perovskite precursor enter the perovskite lattice leads to the diffraction peak of δ -FAPbI₃ shifted to high-angle region of 12.3°. During annealing process from 30s to 5min, the position of the δ -FAPbI₃ peak gradually shifts to low-angle region from 12.3° to 11.9° owing to the substitution of Cl ions by I ions [M. Kim, et al., *Joule*, 2019, **3**, 2179].

As the reviewer's carefully observation, the diffraction peak next to δ -FAPbI₃ (12.3°) is the residual MA₂Pb₃I₈·2DMSO diffraction peak at 11.87°, which is overlapped with δ -FAPbI₃ before using anti-solvent as shown in Supplementary Fig. 5. The XRD data of the intermediate phase MA₂Pb₃I₈·2DMSO is consistent with Ref. [Zhang, K. et al. *Nat. Commun.* **11**, 1006 (2020).]

Supplementary Fig. 5. X-ray diffraction (XRD) patterns of the wet perovskite film a) perovskite film fabricated using perovskite precursors consisting of MAI: PbI₂ (1:1, mol%) in DMF: DMSO (8:1, v/v). b) control perovskite film without anti-solvent during spin-coating, c) control perovskite film with anti-solvent during spin-coating.

We added the related discussion and cited the reference as below.

Changes in the manuscript for *Comment #1 and Comment #2*:

“Therefore, in order to understand the transformation process of the intermediate phases into perovskite, we carried out in-situ tracking of XRD (Fig. 2a) to investigate different processes. For the control FAMA perovskites, although only small amounts of MA⁺ were used in the perovskite precursor, complex intermediate phases, e.g., the solvate phases MA₂Pb₃I₈·2DMSO (2θ= 6.73°, 7.38°, 9.33°) and δ-FAPbI₃ (2θ = 11.9°), were formed from the DMSO/DMF solvent in the wet perovskite film without using anti-solvent. The formation of MA₂Pb₃I₈·2DMSO was due to a strong interaction between perovskite precursor and DMSO, which was confirmed by the XRD pattern of the intermediate phase of MA₂Pb₃I₈·2DMSO (Supplementary Fig. 5) and agreed with previous report²⁶. After using antisolvent to extract DMF/DMSO, most of the intermediate solvate phase transformed to α-phase of perovskite⁹. However, MA₂Pb₃I₈·2DMSO and δ-FAPbI₃ were still present in the film. As the anti-solvent assisted crystallization process^{27,28}, Cl⁻ from MACl in the perovskite precursor enter the perovskite lattice, which leads to the diffraction peak of δ-FAPbI₃ shifted to high-angle region of 12.3°. The diffraction peak next to δ-FAPbI₃ (12.3°) is the residual MA₂Pb₃I₈·2DMSO diffraction peak at 11.87°, which is overlapped with δ-FAPbI₃ before using anti-solvent as shown in the enlarged XRD patterns in Supplementary Figure 5. During annealing process from 30s to 5min, the position of the δ-FAPbI₃ peak (from 12.3° to 11.9°) and α-FAPbI₃ (from 14.15° to 13.9°) gradually shifts to low-angle region owing to the substitution of Cl ions by I ions.^{5,6} Noticeably, δ-FAPbI₃ was consistently observed during α-FAPbI₃ crystallization, which might be due to two competition pathways of crystal-nucleation originating from the intermediate phases as shown below and the schematic diagram is shown in Fig. 1a,

Therefore, it is unfeasible to control the nucleation and crystal growth of the α-FAPbI₃ owing to the complex perovskite intermediates. After annealing at 100°C for 1 hours, the peak of α-phase perovskite dominated, but PbI₂ (2θ =12.8°) was observed due to the thermal decomposition of the unstable non-perovskite yellow phase.”

Comment #3: The authors applied FTPA to the bulk of perovskite films (B-FTPA) and further on the B-FTPA film surface (S-FTPA) to passivate defects, achieving improved device performance. If just applying FTPA onto the control films with bulk passivation, how does the device perform?

Response: We sincerely appreciate the reviewer's suggestion. During our optimization of the perovskite solar cells, we have optimized FTPA based PSCs in three different ways, i) **B-FTPA:** Using FTPA as additive in the bulk of perovskite film, ii) **S-FTPA:** Using FTPA as the surface layer on the perovskite film, iii) **BS-FTPA:** Using optimized FTPA as additive in the bulk of perovskite film, then further using FTPA as the surface layer coated on the B-FTPA film.

Note: *To avoid confusion, we have changed the abbreviation S-FTPA in the original manuscript to BS-FTPA in the revised manuscript.*

During the optimization of the PSCs, we first optimized different concentration of FTPA (2, 5, 10 mol%) as the additive in the control perovskite film. The best photovoltaic performance of B-FTPA PSCs is to use 5 mol% FTPA in the perovskite precursor. Then we optimized different concentration of FTPA (2, 5, 10, 15mg mL⁻¹) as the interlayer on the control perovskite, and we found that 2mg mL⁻¹ FTPA treated PSCs (S-FTPA) exhibited the best photovoltaic performance. Furthermore, we selected the B-FTPA with the optimal doping concentration of 5 mol% FTPA, and optimized different concentration of FTPA (2, 5, 10, 15mg mL⁻¹) as the interlayer, and we found that 2mg mL⁻¹ FTPA treated PSCs (BS-FTPA) exhibited the best photovoltaic performance.

We have added the optimized PSC performance in the Supplementary Fig. 40-42 and Supplementary Table 5-7 as below.

Changes in the Supporting information:

i) B-FTPA: Using FTPA as additive in the bulk of perovskite

Device architecture: FTO/CBD-SnO₂/ FA_{0.95}MA_{0.05}Pb(I_{0.95}Br_{0.05})₃-FTPA/Spiro-OMeTAD/Au

Supplementary Fig. 40. Optimal *J-V* curves of the n-i-p PSCs with different FTPA additive concentration (mol%) in the reverse scan direction.

Supplementary Table 5. The best photovoltaic parameters of n-i-p PSCs for the control and B-FTPA devices with different FTPA concentration measured in reverse scan directions under standard AM 1.5 illumination (100 mW cm⁻²).

	FTPA (mol%)	V_{oc} (V)	J_{sc} (mA cm ⁻²)	FF (%)	PCE (%)
	Control	1.131	24.29	80.11	22.01
B-FTPA	2 (17.25 mg mL ⁻¹)	1.139	24.13	81.08	22.29
	5 (43.13 mg mL ⁻¹)	1.161	24.25	82.41	23.22
	10 (86.26 mg mL ⁻¹)	1.113	24.23	79.64	21.48

ii) S-FTPA: Using FTPA as the surface layer on the perovskite film

Device architecture: FTO/CBD-SnO₂/ FA_{0.95}MA_{0.05}Pb(I_{0.95}Br_{0.05})₃/FTPA/Spiro-OMeTAD/Au

Supplementary Fig. 41. Optimal J - V curves of the n-i-p PSCs with different FTPA interfacial treatment concentration (mg mL⁻¹) in the reverse scan direction.

Supplementary Table 6. The best photovoltaic parameters of PSCs for the n-i-p PSCs with different FTPA concentration measured in reverse scan directions under standard AM 1.5 illumination (100 mW cm⁻²).

	FTPA (mg mL ⁻¹)	V _{oc} (V)	J _{sc} (mA cm ⁻²)	FF (%)	PCE (%)
S-FTPA	2	1.154	24.46	81.68	23.05
	5	1.148	24.44	80.50	22.58
	10	1.136	24.40	80.37	22.27
	15	1.122	24.43	79.91	21.90

iii) BS-FTPA: Using optimized FTPA as additive in the bulk of perovskite films, then further using FTPA as the surface layer coated on the B-FTPA film

Device architecture: FTO/CBD-SnO₂/ FA_{0.95}MA_{0.05}Pb(I_{0.95}Br_{0.05})₃-FTPA/FTPA/Spiro-OMeTAD/Au

Supplementary Fig. 42. Optimal *J-V* curves of the B-FTPA (5mol% FTPA in bulk) based n-i-p PSCs with different FTPA interfacial treatment concentration (mg mL⁻¹) in the reverse scan direction.

Supplementary Table 7. The best photovoltaic parameters of PSCs for the B-FTPA (5mol% FTPA in bulk) based n-i-p PSCs with different FTPA interfacial treatment concentration (mg mL⁻¹) measured in reverse scan directions under standard AM 1.5 illumination (100 mW cm⁻²).

	FTPA (mg mL ⁻¹)	V _{oc} (V)	J _{sc} (mA cm ⁻²)	FF (%)	PCE (%)
BS-FTPA	2	1.182	24.43	83.45	24.10
	5	1.158	24.59	81.84	23.32
	10	1.148	24.45	81.37	22.85
	15	1.143	24.46	80.44	22.49

Comment #4: AIBN was used to facilitate the polymerization process at lower temperature. Is there any residue of AIBN in the final perovskite films?

Response: We sincerely appreciate review's important suggestion. AIBN was used as a free radical initiator in the perovskite precursor to help FTPA convert into cross-linked polymer via atom transfer radical polymerization (ATRP) during perovskite annealing-process. For the B-FTPA system, the molar concentration of perovskite is 1.474mmol, the doping ratio of FTPA was 5 mol% of perovskite about 0.073mmol in the perovskite precursor. AIBN is only 0.1mol% of FTPA about 0.73×10^{-4} mmol in the perovskite precursor, which is one ten thousandth of the perovskite content. Compared to the perovskite precursor and FTPA, the addition of trace amounts of AIBN does not affect the crystallization kinetics of perovskite. In addition, the thermal decomposition temperature of AIBN measured by Thermogravimetric analysis (TGA) (Supplementary Fig. 15b) is between 75 and 85°C to produce free radicals, and then fast decompose after 100 °C (the annealing temperature of perovskite), thus does not remain in the final perovskite films. The differential scanning calorimeter (DSC) (Supplementary Fig. 15c) was also confirmed the thermal process as reported in the Ref. (Y. Zhao et al. Adv. Mater. 2020, 32, 1907769; S. Guo, et al., J. Therm. Anal. Calorim. (2013) 113:1169–1176; X. Li et al. J. Hazard. Mater. 2008, 159, 13.).

Changes in the Supporting information:

Supplementary Figure 15. (a) Differential Scanning Calorimetry (DSC) thermograms of FTPA and FTPA with the free radical initiator of 2,2'-azobis-isobutyronitrile (AIBN, 0.1mol%). (b) TGA curves of AIBN. (c) DSC curves of AIBN.

Note: AIBN was used as a free radical initiator in the perovskite precursor to help FTPA convert into

cross-linked polymer via atom transfer radical polymerization (ATRP) during perovskite annealing-process. For the B-FTPA system, the molar concentration of perovskite is 1.474mmol, the doping ratio of FTPA was 5 mol% of perovskite about 0.073mmol in the perovskite precursor. AIBN is only 0.1mol% of FTPA about 0.73×10^{-4} mmol in the perovskite precursor, which is one ten thousandth of the perovskite content. Compared to the perovskite precursor and FTPA, the addition of trace amounts of AIBN does not affect the crystallization kinetics of perovskite. In addition, the thermal decomposition temperature of AIBN measured by Thermogravimetric analysis (TGA) (Supplementary Fig. 15b) is between 75 and 85°C to produce free radicals, and then fast decompose after 100 °C (the annealing temperature of perovskite), thus does not remain in the final perovskite films. The differential scanning calorimeter (DSC) (Supplementary Fig. 15c) was also confirmed the thermal process as reported in the Ref. [1-3]

[1] Y. Zhao et al. *Adv. Mater.* 2020, **32**, 1907769;

[2] S. Guo, et al., *J. Therm. Anal. Calorim.* 2013, **113**, 1169–1176;

[3] X. Li et al. *J. Hazard. Mater.* 2008, **159**, 13.

Comment #5: Is FTPA a universal additive? Can it enhance the device performance when applied in the inverted (p-i-n) perovskite solar cells?

Response: Thanks for the reviewer's important suggestion about the universal use of FTPA. As the reviewer's suggestion, we have applied FTPA as the perovskite additive in the inverted (p-i-n) perovskite solar cells using two different perovskite precursors, $\text{FA}_{0.95}\text{MA}_{0.05}\text{Pb}(\text{I}_{0.95}\text{Br}_{0.05})_3$ and $\text{Cs}_{0.05}\text{FA}_{0.80}\text{MA}_{0.15}\text{Pb}(\text{I}_{0.85}\text{Br}_{0.15})_3$, respectively. The architecture of the perovskite is ITO/PTAA/perovskite-5mol% FTPA/ $\text{PC}_{61}\text{BM}/\text{C}_{60}/\text{BCP}/\text{Ag}$. We have added the related discussion in the revised manuscript and figures in the Supporting information as below. As shown in (Supplementary Fig. 29), the PCEs of the PSCs with FTPA as the additive were all improved in both perovskite systems. The main increased PSCs parameters are V_{OC} and FF, owing to suppressed non-radiative recombination and the improved charge transport.

Changes in the Supporting Information:

Supplementary Figure 29. *J-V* curves of B-FTPA PSCs using different perovskite precursors, a) $\text{FA}_{0.95}\text{MA}_{0.05}\text{Pb}(\text{I}_{0.95}\text{Br}_{0.05})_3$ and b) $\text{Cs}_{0.05}\text{FA}_{0.80}\text{MA}_{0.15}\text{Pb}(\text{I}_{0.85}\text{Br}_{0.15})_3$ in the reverse scan direction.

Based on the comments of three reviewers, we have added the following discussion of the generality of FTPA in PSCs.

Changes in the manuscript:

“The generality of FTPA strategy

Through investigating the effects of FTPA on perovskite crystallization, energy level modulation, and carrier transport balance, we believe that FTPA should be generally applicable to other cation and halide compositions and different PSC architectures. The corresponding photovoltaic data were summarized in Table 1. We applied FTPA as the perovskite additive in the triplication mixed-halide perovskite system, $\text{Cs}_{0.05}\text{FA}_{0.85}\text{MA}_{0.10}\text{Pb}(\text{I}_{0.97}\text{Br}_{0.03})_3$ (Abbreviated as CsFAMA), and investigated the effect of FTPA on the crystallization kinetics of Cs-containing perovskite system by in-situ XRD, optical microscopy images, SEM images and in-situ UV-vis absorption spectra as shown in Supplementary Fig. 27. In-situ XRD showed that the intermediate phases were also mainly derived from $\text{MA}_2\text{Pb}_3\text{I}_8 \cdot 2\text{DMSO}$ and $\delta\text{-FAPbI}_3$ as the MAFA perovskite system, and the Cs-related intermediate phase $\delta\text{-CsPbI}_3$ ($2\theta = 9.7^\circ$) was not observed.⁴⁵ The addition of FTPA could restrain the intermediate phase and promoted the formation of a better perovskite crystallinity. The optical microscopy images and SEM images also proved that the less intermediate phases during the perovskite formation stage facilitate larger crystal grains of the perovskite films. In-situ UV-vis absorption spectra during spin-coating and annealing process of the perovskite also indicated that the strong internal interactions between the FTPA and perovskite slowed down the crystallization of perovskite, which is essential for the high crystal quality of the treated perovskite. Interestingly, the CsFAMA-perovskite system (5s) began to crystallize earlier than the FAMA-perovskite system (15s) after annealing, probably due to the addition of Cs ions promoting the formation of perovskite crystals. PSC performance based on CsFAMA-perovskite in both n-i-p and p-i-n perovskite architectures were improved compared to the reference sample (Supplementary Fig. 28-29). In addition, benefits from high hole mobility and good energy level alignment with perovskite, we applied FTPA as the dopant-free hole transporting materials (HTM) to alternative the traditional PTAA in p-i-n PSCs,⁴⁶ and the device exhibited a champion PCE of 22.48% compared to the control device (19.94%) (Supplementary Fig. 30 and Supplementary table 3).

Table 1. Device parameters of champion PSCs based on different device architecture and perovskite composition via FTPA modification.

PSC Architecture	Perovskite Composition	FTPA Application	V_{oc} [V]	J_{sc} [mA cm^{-2}]	FF [%]	PCE [%]
n-i-p	$\text{FA}_{0.95}\text{MA}_{0.05}\text{Pb}(\text{I}_{0.95}\text{Br}_{0.05})_3$	Control ^{a)}	1.143	24.46	80.38	22.48
		Bulk ^{b)}	1.161	24.25	82.41	23.22
		Bulk/Surface ^{c)}	1.182	24.43	83.45	24.10
	$\text{Cs}_{0.05}\text{FA}_{0.85}\text{MA}_{0.10}\text{Pb}(\text{I}_{0.97}\text{Br}_{0.03})_3$	Control ^{a)}	1.157	23.95	76.78	21.27
		Bulk ^{b)}	1.174	24.02	79.85	22.52
		Bulk/Surface ^{c)}	1.196	24.11	80.77	23.29
p-i-n	$\text{FA}_{0.95}\text{MA}_{0.05}\text{Pb}(\text{I}_{0.95}\text{Br}_{0.05})_3$	Control ^{d)}	1.031	23.01	79.06	18.75
		Bulk ^{e)}	1.126	23.22	82.09	21.46
	$\text{Cs}_{0.05}\text{FA}_{0.80}\text{MA}_{0.15}\text{Pb}(\text{I}_{0.85}\text{Br}_{0.15})_3$	Control ^{d)}	1.094	23.48	79.88	20.51
		Bulk ^{e)}	1.138	23.74	81.54	22.03
	$\text{Cs}_{0.05}\text{FA}_{0.80}\text{MA}_{0.15}\text{Pb}(\text{I}_{0.85}\text{Br}_{0.15})_3$	Control ^{f)}	1.081	23.39	78.89	19.94
		HTL ^{g)}	1.156	23.93	81.27	22.48

a) FTO/CBD-SnO₂/perovskite/Spiro-OMeTAD/Au; b) FTO/CBD-SnO₂/perovskite-FTPA/Spiro-OMeTAD/Au; c) FTO/CBD-SnO₂/perovskite-FTPA/FTPA/Spiro-OMeTAD/Au; d) ITO/PTAA/perovskite/PC₆₁BM/C₆₀/BCP/Ag; e) ITO/PTAA/perovskite-FTPA/PC₆₁BM/C₆₀/BCP/Ag; f) ITO/PTAA/perovskite/PC₆₁BM/C₆₀/BCP/Ag; g) ITO/FTPA/perovskite/PC₆₁BM/C₆₀/BCP/Ag.

Comment #6: When referring to the illumination stability, please add the test temperature of the samples. In addition to the light and humidity stability, how about the thermal stability of the B-FTPA and S-FTPA samples?

Response: We sincerely appreciate review's important suggestion. For the illumination stability, the test temperature of the PSCs is about 23 ± 2 °C aged in a homemade thermostats instrument combining a thermoelectric cooler, objective table, light source and temperature display as shown in Supplementary Fig. 25. As the reviewer's suggestion, for the thermal stability, we tracked the morphological degradation of the perovskite layers under 85 °C for 500 hours as shown in Supplementary Fig. 26. We also measured the thermal stability of the control, B-FTPA and BS-FTPA PSCs under 65 °C on a hot plate under a nitrogen atmosphere (ISOS-T-1) for 500 hours as shown in Figure 4.f

Changes in the manuscript:

“Devices were aged in a thermostats (23 ± 2 °C, Supplementary Fig. 25) at simulated 1-sun illumination under a nitrogen atmosphere (ISOS-LC-1) (Fig. 4e).”

Changes in the Supporting Information:

Supplementary Figure 25. For the illumination stability, the test temperature of the PSCs is about 23 ± 2 °C aged in a homemade thermostats instrument combining a thermoelectric cooler, objective table, light source, and temperature display.

Changes in the manuscript:

“We have tracked the XRD patterns to estimate the morphological degradation of the control, B-FTPA and BS-FTPA perovskite films under 85 °C (Supplementary Fig. 26). The control film exhibited obvious deterioration after 500 hours manifested by a significant increase of the PbI_2 peak, whereas the XRD pattern of the B-FTPA film showed only few of PbI_2 peak, and no apparent impurity peaks were observed in the BS-FTPA film. In addition, the long-term stability of the unencapsulated PSCs at temperature of 65 °C under a nitrogen atmosphere (ISOS-T-1, Figure 4i) were also examined. After 500 hours aging, B-FTA ad BS-FTPA device stabilized over 80% and 85% of initial efficiency, while the control device only retains 58% of its initial efficiency merely after 500 h. The better thermal stability of FTPA-based PSCs might be attributed to the inhibition of ion migration by the polymer network formed in perovskite film.⁴⁴”

Supplementary Fig. 26. Time evolution of XRD patterns of the Control, B-FTPA and BS-FTPA perovskite films at temperature of 85 °C under a nitrogen atmosphere.

Figure 4 f, Device stability of unencapsulated devices under 1-sun illumination at 23±2 °C in a nitrogen atmosphere (ISOS-LC-1). **g**, Device stability of unencapsulated devices held at 25±5 °C and 50±10% relative humidity (ISOS-D-1). **i**, Device stability of unencapsulated devices under 65±3°C thermal aging (ISOS-T-1). All of the error bars represent the standard deviation for six devices.

Reply to reviewer #3

In this manuscript, the authors have chosen a very important topic and have focused on the control of crystallization of the FA-based perovskites. The topic area of the work is timely and interesting. The authors show very convincingly that the mixed halide perovskite generally fails to form high-quality film, due to the multiple pathways of crystal nucleation originating from various intermediate phases in the film-forming process. Series of in-situ evidence such as XRD, UV-vis and GIWAXS provide a deep insight into the intermediate phase, nucleation, and crystallization process of the perovskite films during spin-coating and annealing procedures. To overcome the dilemma, the authors ingeniously design a novel multifunctional fluorinated additive and propose a single crystallization pathway approach to further promote the orientated crystallization of α -FAPbI₃ by the hydrogen-bonded polymer network. An impressive efficiency of perovskite solar cells above 24% is obtained with high stability, even the bare perovskite film can against liquid water long periods of time. The rationalized design of the efficient molecular additive in this study could motivate similar work for efficient and stable FAPbI₃ based PSCs.

The analysis in this work is very thorough. The experimental data are comprehensive and credible, which strongly support the conclusion. In summary, I think this work is important and innovative for the field of perovskite solar cells and will be of wide interest to the readers. I strongly recommend publication of this study in Nature Communication.

General Response: We are grateful to the reviewer for the nice comments and well appreciating the importance of our work. We carefully considered these comments and the detailed response can be found in the point-to-point response below.

Comment #1: As shown in spectra of XPS (in Fig. 1d and Supplementary Fig. 4), to characterize the interaction of the additive with perovskite, the elemental F region are inadequate and the authors should provide the XPS spectrum of the intrinsic FTPA to fully demonstrate the interaction.

Response: Thanks for the reviewer's important suggestion. As suggested by the reviewer, we have added the related discussion in the revised manuscript and XPS spectrum of the intrinsic FTPA in the (Supplementary Fig. 4) as below.

Changes in the manuscript: "As shown in Fig. 1d, XPS spectral profile showed that the main peaks at 143.15 (Pb 4f_{5/2}) and 138.25 (Pb 4f_{7/2}) eV were shifted toward low binding energy regions (142.70 and 137.80 eV, respectively) after the introduction of FTPA in the perovskite, whereas the F 1s peak of perovskite with FTPA showed the opposite trend (Supplementary Fig. 4). This could be attributed to the interaction between F group of FTPA and uncoordinated Pb ion."

Changes in the Supporting Information:

Supplementary Fig. 4. XPS spectra of pure FTPA, the control perovskite films and perovskite films prepared with FTPA as additive. (a) Pb 4f, (b) I 3d, (c) F 1s, (d) C 1s.

Comment #2: FTPA is used as an additive in the perovskite precursor to induce crystallization and stabilize the α -phase of FAPbI₃. Whether FTPA affects the perovskite lattice and alters the perovskite band gap is something the authors need to consider.

Response: We sincerely appreciate the reviewer's comment. We measured the UV absorption of the perovskite film and found that the perovskite band gap was not changed at 1.55 eV after addition of FTPA compared to the control film as shown in (Supplementary Fig.21).

Changes in the Supporting Information:

Supplementary Fig. 21. UV-vis absorption spectra of the Control, B-FTPA and BS-FTPA perovskite films.

Comment #3: SCLC measurements in Fig. 3c and 3d exhibit the FTPA improved the hole charge transport in the doped perovskite owing to the intrinsic high hole-transport mobility, which contribute to the balance transport of electrons and holes in perovskite devices. And Fig. 3f and 3g also show that FTPA has an aligned energy-level with perovskite as the interfacial hole-transport layer. I am interested in whether the authors have considered the application of FTPA as a hole transport layer to p-i-n structured PSCs as a substitute for the classical PTAA.

Response: We appreciate to the reviewer's important suggestion about the universal use of FTPA. We absolutely agree that FTPA could be used as hole transporting layer (HTL) owing to the high hole-transport mobility and aligned energy-level with perovskite. As the reviewer's suggestion, we have applied FTPA as HTL in the inverted (p-i-n) perovskite solar cells. The architecture of the perovskite solar cell is ITO/FTPA/ $\text{Cs}_{0.05}\text{FA}_{0.80}\text{MA}_{0.15}\text{Pb}(\text{I}_{0.85}\text{Br}_{0.15})_3$ /PC₆₁BM/C₆₀/BCP/Ag. As shown in Supplementary Fig. 30 and Supplementary Table 3, the optimized PSCs using FTPA as HTL (4mg/mL) exhibited an excellent PCE of 22.48% with V_{OC} of 1.156 V, J_{sc} of 23.93 mA cm⁻² and a FF of 81.27% compared to the traditional HTL PTAA (PCE=19.94%). We have added the related discussion in the revised manuscript and figures in the Supplementary as below.

Changes in the Supporting Information:

Supplementary Fig. 30. Optimal J - V curves of the PSCs measured in the reverse scan direction using different concentration of FTPA as HTL and PTAA as reference. The architecture of the PSCs is ITO/FTPA/ $\text{Cs}_{0.05}\text{FA}_{0.80}\text{MA}_{0.15}\text{Pb}(\text{I}_{0.85}\text{Br}_{0.15})_3$ /PC₆₁BM/C₆₀/BCP/Ag.

Supplementary Table 3. The best photovoltaic parameters of PSCs using different concentration of FTPA as HTL and PTAA as reference measured in reverse scan directions under standard AM 1.5 illumination (100 mW cm^{-2}).

	Concentration (mg mL^{-1})	V_{oc} (V)	J_{sc} (mA cm^{-2})	FF (%)	PCE (%)
PTAA	4	1.081	23.39	78.89	19.95
FTPA	2	1.130	23.70	80.41	21.53
	4	1.156	23.93	81.27	22.48
	8	1.122	23.71	80.21	21.34
	10	1.101	23.78	80.03	20.95

Based on the comments of three reviewers, we have added the following discussion of the generality of FTPA in PSCs.

“The generality of FTPA strategy

Through investigating the effects of FTPA on perovskite crystallization, energy level modulation, and carrier transport balance, we believe that FTPA should be generally applicable to other cation and halide compositions and different PSC architectures. The corresponding photovoltaic data were summarized in Table 1. We applied FTPA as the perovskite additive in the triplication mixed-halide perovskite system, $\text{Cs}_{0.05}\text{FA}_{0.85}\text{MA}_{0.10}\text{Pb}(\text{I}_{0.97}\text{Br}_{0.03})_3$ (Abbreviated as CsFAMA), and investigated the effect of FTPA on the crystallization kinetics of Cs-containing perovskite system by in-situ XRD, optical microscopy images, SEM images and in-situ UV-vis absorption spectra as shown in Supplementary Fig. 27. In-situ XRD showed that the intermediate phases were also mainly derived from $\text{MA}_2\text{Pb}_3\text{I}_8 \cdot 2\text{DMSO}$ and $\delta\text{-FAPbI}_3$ as the MAFA perovskite system, and the Cs-related intermediate phase $\delta\text{-CsPbI}_3$ ($2\theta = 9.7^\circ$) was not observed.⁴⁵ The addition of FTPA could restrain the intermediate phase and promoted the formation of a better perovskite crystallinity. The optical microscopy images and SEM images also proved that the less intermediate phases during the perovskite formation stage facilitate larger crystal grains of the perovskite films. In-situ UV-vis absorption spectra during spin-coating and annealing process of the perovskite also indicated that the strong internal interactions between the FTPA and perovskite slowed down the crystallization of

perovskite, which is essential for the high crystal quality of the treated perovskite. Interestingly, the CsFAMA-perovskite system (5s) began to crystallize earlier than the FAMA-perovskite system (15s) after annealing, probably due to the addition of Cs ions promoting the formation of perovskite crystals. PSC performance based on CsFAMA-perovskite in both n-i-p and p-i-n perovskite architectures were improved compared to the reference sample (Supplementary Fig. 28-29). In addition, benefits from high hole mobility and good energy level alignment with perovskite, we applied FTPA as the dopant-free hole transporting materials (HTM) to alternative the traditional PTAA in p-i-n PSCs,⁴⁶ and the device exhibited a champion PCE of 22.48% compared to the control device (19.94%) (Supplementary Fig. 30 and Supplementary table 3).

Table 1. Device parameters of champion PSCs based on different device architecture and perovskite composition via FTPA modification.

PSC Architecture	Perovskite Composition	FTPA Application	V_{oc} [V]	J_{sc} [mA cm^{-2}]	FF [%]	PCE [%]
n-i-p	$\text{FA}_{0.95}\text{MA}_{0.05}\text{Pb}(\text{I}_{0.95}\text{Br}_{0.05})_3$	Control ^{a)}	1.143	24.46	80.38	22.48
		Bulk ^{b)}	1.161	24.25	82.41	23.22
		Bulk/Surface ^{c)}	1.182	24.43	83.45	24.10
	$\text{Cs}_{0.05}\text{FA}_{0.85}\text{MA}_{0.10}\text{Pb}(\text{I}_{0.97}\text{Br}_{0.03})_3$	Control ^{a)}	1.157	23.95	76.78	21.27
		Bulk ^{b)}	1.174	24.02	79.85	22.52
		Bulk/Surface ^{c)}	1.196	24.11	80.77	23.29
p-i-n	$\text{FA}_{0.95}\text{MA}_{0.05}\text{Pb}(\text{I}_{0.95}\text{Br}_{0.05})_3$	Control ^{d)}	1.031	23.01	79.06	18.75
		Bulk ^{e)}	1.126	23.22	82.09	21.46
	$\text{Cs}_{0.05}\text{FA}_{0.80}\text{MA}_{0.15}\text{Pb}(\text{I}_{0.85}\text{Br}_{0.15})_3$	Control ^{d)}	1.094	23.48	79.88	20.51
		Bulk ^{e)}	1.138	23.74	81.54	22.03
	$\text{Cs}_{0.05}\text{FA}_{0.80}\text{MA}_{0.15}\text{Pb}(\text{I}_{0.85}\text{Br}_{0.15})_3$	Control ^{f)}	1.081	23.39	78.89	19.94
		HTL ^{g)}	1.156	23.93	81.27	22.48

a) FTO/CBD-SnO₂/perovskite/Spiro-OMeTAD/Au; b) FTO/CBD-SnO₂/perovskite-FTPA/Spiro-OMeTAD/Au; c) FTO/CBD-SnO₂/perovskite-FTPA/FTPA/Spiro-OMeTAD/Au; d) ITO/PTAA/perovskite/PC₆₁BM/C₆₀/BCP/Ag; e) ITO/PTAA/perovskite-FTPA/PC₆₁BM/C₆₀/BCP/Ag; f) ITO/PTAA/perovskite/PC₆₁BM/C₆₀/BCP/Ag; g) ITO/FTPA/perovskite/PC₆₁BM/C₆₀/BCP/Ag.

REVIEWERS' COMMENTS

Reviewer #1 (Remarks to the Author):

Thanks for addressing my comments. The revised version can be published.

Reviewer #2 (Remarks to the Author):

The revised manuscript has addressed all the proposed comments and performed additional experiments for the proof. Some supporting data have been also added for a thorough understanding: for example, the XRD peak shift explanation, the thermal stability data and passivation comparison. In my opinion, the revised manuscript looks much improved at this stage and can be published in Nature Communications.

Reviewer #3 (Remarks to the Author):

The revision is acceptable to me.